# State Space Model with Continuous Limit of HiPPO Matrix: Eigenvalue Analysis and Explicit Solution Formula

**Atsushi Takabatake** [1] [2]    **Takaharu Yaguchi** [1] [2]

## Abstract

As a lightweight model for sequence processing, an LSSL that uses the HiPPO matrix has been proposed. In this paper, as the continuous limit of the HiPPO matrix, we propose a Continuized-HiPPO Operator on a function space. Furthermore, as examples of advantages obtained by using this operator, we show that one can analyze the asymptotic behavior of eigenvalues of matrices related to the HiPPO matrix, and that the LSSL induced by the continuous limit Continuized-HiPPO Operator admits an explicit solution.

## 1. Introduction

In deep learning architectures, various approaches have been developed, such as CNNs using convolutions and ResNets employing skip connections. Focusing on time-series data, notable architectures include LSTMs (Hochreiter & Schmidhuber, 1997), RNNs using seq2seq models (Sutskever et al., 2014), and Transformers (Vaswani et al., 2017) utilizing attention layers, which have become the most prominent approach in recent years. Although the architectures mentioned above were proposed in a discrete form, there are also many studies that consider their continuous limits. For example, in Neural-ODE (Chen et al., 2018), the architecture for infinitely deep layers is expressed as an integral, meaning it considers the continuous limit in the direction of the layer. In Neural CDE (Kidger et al., 2020), it considers the continuous limit in time direction.

Recently, architectures based on State Space Models (SSMs) have also been proposed. In these architectures, there are Mamba (Gu & Dao, 2024), Hyena (Poli et al.,

2023), Liquid S4 (Hasani et al., 2023) as examples, especially, these are based on the Linear State Space Layer (LSSL) architecture (Gu et al., 2021). In this paper, we focus on this LSSL-based architecture for handling time-series data.

In the initialization of matrices used for LSSL, specific matrices have been proposed. Specifically, one such matrix is the HiPPO (High-order Polynomial Projection Operators) matrix (Gu et al., 2020). This initialization is known to improve LSSL performance and has been adopted in multiple models, including the S4 model (Gu et al., 2022b), the evolved S4-Diagonal model, and the S5 model (Smith et al., 2023). Among state-space models, the S4D model (Gupta et al., 2022) aims to achieve faster computation by imposing a diagonal structure on the internal matrices. However, it is known that directly using the diagonalization of the HiPPO matrix for initialization is numerically unstable (Gu et al., 2022b). Instead of diagonalizing the full HiPPO matrix, it has been proposed to initialize the model using the diagonalization of the normal (regular) part of the HiPPO matrix. In Gu et al. (2022a), several initialization schemes are proposed based on a conjecture regarding the asymptotic behavior of the eigenvalues of this normal part; however, it should be noted that no proof of the conjecture is provided. We prove this conjecture in this paper.

Our proof is based on introducing a continuous limit of the HiPPO matrices, formulated as an operator on an infinite-dimensional vector space, which we call Continuized-HiPPO. In addition, we show that this continuous limit provides not only a proof of that conjecture, but also an interesting result: an explicit solution formula for SSM with Continuized-HiPPO. This explicit solution formula may provide new computational methods and insights for an architecture.

**Contributions**    In this paper,

- ! & We propose Continuized-HiPPO Operator as a continuous limit that maps the HiPPO matrix to an operator on the function space $L^2$.

As examples of the advantages obtained by interpreting the

[1]Kyushu University [2]RIKEN AIP. Correspondence to: Atsushi Takabatake <takabatake.atsushi@riken.jp>, Takaharu Yaguchi <yaguchi@imi.kyushu-u.ac.jp>.

*Proceedings of the 43$^{rd}$ International Conference on Machine Learning*, Seoul, South Korea. PMLR 306, 2026. Copyright 2026 by the author(s).

HiPPO Matrix as an operator, we describe

! & that one can obtain an answer to a conjecture related to the HiPPO matrix (Gu et al., 2022a);

! & that Continuized-HiPPO Operator provides an explicit solution for LSSL.

## 2. Background

### 2.1. LSSL: Linear State Space Layer

A linear state space model (SSM) is defined as follows. (Baras et al., 1974; Gu et al., 2020)

**Definition 2.1.** *Let* $\mathbb{B}_{\text{in}}, \mathbb{B}_{\text{state}}, \mathbb{B}_{\text{out}}$ *be Banach spaces and* $A \in \mathcal{L}\left(\mathbb{B}_{\text{state}}, \mathbb{B}_{\text{state}}\right), B \in \mathcal{L}\left(\mathbb{B}_{\text{in}}, \mathbb{B}_{\text{state}}\right), C \in \mathcal{L}\left(\mathbb{B}_{\text{state}}, \mathbb{B}_{\text{out}}\right), D \in \mathcal{L}\left(\mathbb{B}_{\text{in}}, \mathbb{B}_{\text{out}}\right).$ *The transformation* $\Phi : \text{Dom}\left(\Phi\right) \to \mathbb{B}_{\text{out}}^{[0,T]}$ *represented by*

$$\Phi(u) = Ch + Du, \quad \frac{\partial h}{\partial t} = Ah + Bu \qquad (1)$$

*is called a (Linear time invariant) State Space Model, where* $h$ *denotes the solution of the differential equation for a given input* $u$, *and* $\text{Dom}\left(\Phi\right) \subset \mathbb{B}_{\text{in}}^{[0,T]}$ *holds.*

In this equation, taking Euclidean space as a Banach space and discretizing it in the time $t$ direction is used as one of the layers in deep learning, referred to as a Linear State Space Layer (LSSL)(Gu et al., 2021).

LSSL has Single Input Single Output (SISO) and Multi Input Multi Output (MIMO) types. In SISO case, using $\mathbb{B}_{\text{in}} = \mathbb{R}, \mathbb{B}_{\text{state}} = \mathbb{R}^N, \mathbb{B}_{\text{out}} = \mathbb{R}$ in parallel, the input function can be taken to be one-dimensional. In this paper we restrict to the SISO case. However, for the MIMO case, the input function is merely vectorized, and results analogous to those in this paper can be obtained.

### 2.2. Implicit State-Space's Solution

An LSSL on a Hilbert space is described using the solution of a differential equation. Applying a standard result from semigroup theory to this solution yields the following.

**Lemma 2.2.** *Consider the state-space differential equation on a Hilbert space*

$$\frac{\partial h}{\partial t} = Ah + Bu \qquad (2)$$

*If* $u$ *is Lipschitz continuous on* $t \in [0, T]$ *and* $A, B$ *are bounded linear operators, then a unique strong solution exists and has the following representation.*

$$h(t) = \int_0^t \exp\left((t-s)\mathcal{A}\right)\mathcal{B}[u](s)\text{d}s \qquad (3)$$

*Proof.* See Appendix C for the proof.

Note that this guarantees that a result that held for differential equations on finite-dimensional spaces also holds on a Hilbert space. When the input function is one-dimensional, this implies that the input-output relation of LSSL admits the following convolutional representation:

$$\Phi(u)(t) = \int_0^t k(t-s)u(s)\text{d}s \qquad (4)$$

$$k(t) = \mathcal{C}\exp\left(t\mathcal{A}\right)\mathcal{B}[1], \qquad (5)$$

This representation uses the operator exponential $\exp\left((t-s)\mathcal{A}\right)$, and in this sense it is implicit. Here, we call $k$ the SSM convolutional kernel.

### 2.3. HiPPO: High-order Polynomial Projection Operators

Given a function $f : s \mapsto f(s)$, we consider approximating it online with respect to time $t$ using an orthonormal system $\{p_n : (t,s) \mapsto p_n(t,s)\}_{n \in \mathbb{N}}$ and a measure $\omega : (t,s) \mapsto \omega(t,s)$. In this case, it is known that the coefficients collected as a finite-dimensional vector $h(t)$ can be represented by a linear ODE on a finite-dimensional space (Gu et al., 2023). In particular, by choosing the pair of measure and basis as

$$\omega(t,s) = e^{-(t-s)}\mathbb{1}_{[-\infty,t)}, p_n(t,s) = L_n\left(e^{-(t-s)}\right), \quad (6)$$

one can represent the coefficients $h(t)$ by the following linear time-invariant ODE.

$$\frac{\text{d}h}{\text{d}t} = A^{\text{hippo}}h + B^{\text{hippo}}u \qquad (7)$$

$$A_{ij}^{\text{hippo}} = -\begin{cases} \sqrt{2i+1}\sqrt{2j+1} & (i > j) \\ i+1 & (i = j) \\ 0 & (i < j) \end{cases} \qquad (8)$$

$$B_{i1}^{\text{hippo}} = \sqrt{2i+1} \qquad (9)$$

Here, $L_n$ denotes the Legendre polynomial, and $i, j = 0, ..., N-1$.

In particular, these matrices are called the HiPPO matrices (Gu et al., 2020; 2022b).

**Definition 2.3.** $A^{\text{hippo}} \in \mathbb{R}^{N \times N}, B^{\text{hippo}} \in \mathbb{R}^{N \times 1}$ *are called the HiPPO (High-order Polynomial Projection Operators) matrices.*

It is (experimentally) known that using this HiPPO matrices as $A, B$ of LSSL ! =either as an initial value of the parameters or as a fixed value ! =improves the performance of LSSL (Gu et al., 2022b).

### 2.4. Eigenvalues of Normal part of HiPPO Matrix

From the viewpoint of computational speed, one sometimes assumes that $A$ in LSSL is diagonal. This is called a

**Diagonal State Space Model.** In this case, initialization by diagonalizing the HiPPO matrix above is numerically unstable (Gu et al., 2022b), so instead it has been proposed to use an initialization based on diagonalizing the normal part of the HiPPO matrix defined below (Gupta et al., 2022).

**Definition 2.4.** *The following $A^{\text{skew}}, A^{\text{lowrank}}$ are called the skew-symmetric part and the low-rank part of the HiPPO matrix.*

$$A_{ij}^{\text{skew}} = -\frac{1}{2} \begin{cases} \sqrt{2i+1}\sqrt{2j+1} & (i > j) \\ 0 & (i = j) \\ -\sqrt{2i+1}\sqrt{2j+1} & (i < j) \end{cases} \quad (10)$$

$$A_{ij}^{\text{lowrank}} = -\frac{1}{2}\sqrt{2i+1}\sqrt{2j+1} \quad (11)$$

*Moreover, the regular matrix $A^{\text{normal}} = A^{\text{skew}} - \frac{1}{2}I$ is called the regular part of the HiPPO matrix.*

This choice makes the original HiPPO and the above SSM kernel $K(t)$ equal in the limit $N \to \infty$ (Gu et al., 2022a; Smith et al., 2023). Note that the HiPPO matrix can be decomposed as

$$A^{\text{hippo}} = A^{\text{normal}} + A^{\text{lowrank}}. \quad (12)$$

In this context, Gu et al. (2022a) considers diagonalizing the normal part

$$A^{\text{diag}} = \text{diag}(A^{\text{normal}}). \quad (13)$$

and makes the following conjecture about the eigenvalues of $A^{\text{normal}}$.

**Conjecture 2.5.** *Assume that the positive imaginary parts $\{\tau_n^{(N)}\}_{n=1,\ldots,\lfloor \frac{N}{2} \rfloor}$ of the eigenvalues of the $N \times N$ matrix $A^{\text{normal}} \in \mathbb{R}^{N \times N}$ are arranged in monotonically decreasing order in $n$. Then, using $c \approx 0.5236$,*

$$\tau_1^{(N)} \approx \frac{N^2}{\pi} + c \quad (14)$$

*holds. Moreover, for each fixed $N$, the imaginary parts of the eigenvalues $\tau_n^{(N)}$ decay on the scale of $\Theta(n^{-1})$.*

In this paper we provide an answer to this conjecture.

### 2.5. Separable Volterra Integral Operator

In operator theory, the following integral operators are considered as typical linear operators on function spaces.

**Definition 2.6.** *An operator $\mathcal{K}$ on $L^2(E)$ that can be written as follows is called an integral operator.*

$$\mathcal{K}[f] = \int_E K(x,\xi)f(\xi)\mathrm{d}\xi \quad (15)$$

*In this case, $K$ is called the integral kernel.*

In what follows, we consider $E = [0, N]$. As an example of an integral operator, the following operator is known.

**Definition 2.7.** *An integral operator over a triangular domain*

$$\mathcal{K}[f] = \int_0^x K(x,\xi)f(\xi)\mathrm{d}\xi \quad (16)$$

*is called a Volterra integral operator. In particular, if the kernel can be written using $\varphi, \psi \in L^2$*

$$K(x,\xi) = \varphi(x)\psi(\xi) \quad (17)$$

*then this kernel is said to be separable. In this case,*

$$\mathcal{K}_{\varphi,\psi}[f] = \varphi(x) \int_0^x \psi(\xi)f(\xi)\mathrm{d}\xi \quad (18)$$

*the operator $\mathcal{K}_{\varphi,\psi}$ represented as above is called a separable Volterra integral operator (Gustavson & Rosen, 2023).*

These are obtained by viewing the integral

$$\int_0^x f(x)\mathrm{d}x \quad (19)$$

as an operator acting on $f$ and generalizing it by introducing an integral kernel. Note that in some references, $K(x,\xi) = \sum_{i=0}^{M} \varphi_i(x)\psi_i(\xi)$ is treated as a separable kernel, but here we consider only $M = 1$.

## 3. Related Work

This study views the Linear State Space Layer (LSSL) based on the HiPPO matrix, from the viewpoint of continuizing it as an operator on a function space, and is related to lines of research spanning existing State Space Models, continuous-time neural networks, and the theory of integral operators. Below, we organize prior work that is particularly closely related to this study.

### 3.1. State Space Models for Long-Range Sequence Modeling

In recent years, as a method for efficiently handling long-range sequences, architectures based on State Space Models (SSMs) that can be computed in linear time have attracted considerable attention. Linear State Space Layer (Gu et al., 2021) was introduced as a framework that unifies recurrent models, convolutional models, and continuous-time models, and later, in the Structured State Space (Gu et al., 2022b) model, it was shown that fast kernel computation is possible by leveraging the Normal-plus-Low-Rank (NPLR/DPLR) structure.

In a direction that emphasizes computational efficiency even further, Diagonal State Space Models that use diagonal or approximately diagonal state-space matrices have

been proposed (Gupta et al., 2022; Gu et al., 2022a), and initialization based on diagonalizing the regular part (normal part) of the HiPPO matrix has been reported to be effective in practice. On the other hand, in these works, discussion of the asymptotic behavior of eigenvalues of the HiPPO matrix and its regular part has mainly been based on empirical and numerical observations, and a rigorous analysis had not been provided.

This study can be positioned within this line of SSM-based sequence models built on HiPPO, but rather than treating HiPPO directly as a discrete matrix, we reformulate it as a continuized operator, with the aim of theoretically deriving eigenvalue asymptotics and explicit kernel representations.

### 3.2. Memory Representations Based on HiPPO and Orthogonal Polynomials

HiPPO (High-order Polynomial Projection Operators) (Gu et al., 2020) was introduced based on the principle of online projection of continuous-time signals onto an orthogonal polynomial basis, as a method that provides an optimal memory representation. In particular, HiPPO-LegS using Legendre polynomials is widely used as a matrix initialization in state space models.

Subsequently, frameworks that generalize the choice of orthogonal bases and measures were proposed, expanding the learnability and design space of SSMs using HiPPO. Moreover, continuous-time memory representations using orthogonal polynomials, such as the Legendre Memory Unit (Voelker et al., 2019), are conceptually close to HiPPO.

### 3.3. Continuous-Time Limits and an Operator-Theoretic Viewpoint

Research that views deep-learning models as continuous-time limits has developed in the form of continuizing either the layer direction or the time direction, for example, Neural ODE (Chen et al., 2018) and Neural CDE (Kidger et al., 2020). In addition, in work represented by Neural Operator (Kovachki et al., 2023), a viewpoint is adopted in which inputs and outputs are regarded as function spaces and learning problems are formulated as infinite-dimensional operators.

This study is also situated in this trend of "continuization", but compared with other work, it differs in that it focuses on linear state space models based on HiPPO and continuizes along the state-space direction. In particular, the Continuized-HiPPO Operator introduced in this study arises naturally as a continuous limit of the HiPPO matrix in the spatial (state) direction, and enables explicit treatment of eigenvalue behavior that is hard to see in the discrete model.

### 3.4. Volterra Integral Operators and Explicit Kernel Representations

Continuized-HiPPO Operator is essentially represented as a Volterra-type integral operator. The theory of Volterra integral equations and the structure of their kernels has been widely studied in classical integral equation theory.

In this study, after describing why this Volterra integral operator admits an explicit solution, we show that the convolution kernel of the LSSL based on the Continuized-HiPPO Operator can be written explicitly as an integral representation using Bessel functions. Such explicit solutions are expected to be useful for theoretical analysis and numerical understanding of SSMs.

As stated above, this study bridges SSM research based on HiPPO, continuous-time models, and the theory of integral operators, and examines the continuous limit of the HiPPO matrix and its properties.

## 4. Continuized-HiPPO Operator

### 4.1. Definition of Continuized-HiPPO Operator

Note that $A^{\text{hippo}}$ is a lower-triangular matrix; thus it can be viewed as a discretization of an integral operator. The theorem stated above is the following.

**Theorem 4.1.** *The (finite-dimensional) differential equation of LSSL using the HiPPO matrices*

$$\frac{\partial h}{\partial t} = A^{\text{hippo}}h + B^{\text{hippo}}u, h(0) = 0 \qquad (20)$$

*coincides with the differential equation on a function space*

$$\frac{\partial \tilde{h}}{\partial t}(t)(x) = -\sqrt{2x+1}\int_0^x \sqrt{2\xi+1}\tilde{h}(t)(\xi)d\xi - \frac{1}{2}\tilde{h}(t)(x)$$
$$+ \sqrt{2x+1}\left(-\frac{1}{2}\int_0^t e^{-(t-s)}u(s)ds + u(t)\right) \qquad (21)$$

*which coincides with the discretization in the $\xi$-direction of (21) by the trapezoidal rule. Here we take the step size as $d\xi \approx \Delta\xi = 1$. That is, $h_i \approx \tilde{h}(i)$.*

*Proof.* See Appendix D for the proof.

Note that when the state dimension $N$ is sufficiently large, $1 = \Delta x \ll N$; hence a discretization width $\Delta x = 1$ can be interpreted as small. This differential equation can be regarded as a State Space Model on the $L^2$ space. Concretely, define the operators $\mathcal{A}^{\text{hippo}} : L^2([0, N], \mathbb{C}) \to$

$L^2([0, N], \mathbb{C})$ and $\mathcal{B}^{\text{hippo}} : \mathbb{C} \to L^2([0, N], \mathbb{C})$ by

$$\mathcal{A}^{\text{hippo}}[f](x) = -\sqrt{2x+1} \int_0^x \sqrt{2\xi+1} f(\xi) d\xi - \frac{1}{2} f(x) \tag{22}$$

$$\mathcal{B}^{\text{hippo}}[a](x) = a\sqrt{2x+1} \tag{23}$$

and redefine the input function as

$$v(t) = -\frac{1}{2} \int_0^t e^{-(t-s)} u(s) ds + u(t) \tag{24}$$

then (21) can be written as follows.

$$\frac{\partial \tilde{h}}{\partial t}(t) = \mathcal{A}^{\text{hippo}}[\tilde{h}(t)] + \mathcal{B}^{\text{hippo}}[v(t)] \tag{25}$$

In particular, this is exactly the state space model on the infinite-dimensional space $L^2([0, N])$. Therefore, we define these $\mathcal{A}^{\text{hippo}}, \mathcal{B}^{\text{hippo}}$ as the definition of Continuized-HiPPO Operator.

**Definition 4.2.** *We call* $\mathcal{A}^{\text{hippo}}, \mathcal{B}^{\text{hippo}}$ *Continuized-HiPPO Operators (CHIP Operator).*

This can be interpreted as taking the continuous limit of the HiPPO matrix in the spatial direction, or as capturing the global behavior of the HiPPO matrix when the state dimension $N$ is large. In particular, in Section 5 we use this interpretation to answer Conjecture 2.5.

### 4.2. Decomposition of Continuized-HiPPO Operator

As in the finite-dimensional case, $\mathcal{A}^{\text{hippo}}$ can be decomposed as follows.

**Definition 4.3.** *When we decompose Continuized-HiPPO Operator as follows, we call* $\mathcal{A}^{\text{skew}}$ *the skew-symmetric part of* $\mathcal{A}^{\text{hippo}}$, $\mathcal{A}^{\text{lowrank}}$ *the low-rank part of* $\mathcal{A}^{\text{hippo}}$, *and* $\mathcal{A}^{\text{skew}} - \frac{1}{2}\mathcal{I}$ *the normal part of* $\mathcal{A}^{\text{hippo}}$.

$$\mathcal{A}^{\text{hippo}} = \mathcal{A}^{\text{skew}} - \frac{1}{2}\mathcal{I} + \mathcal{A}^{\text{lowrank}} \tag{26}$$

$$\mathcal{A}^{\text{skew}}[f](x) = -\frac{1}{2}\left( \sqrt{2x+1} \int_0^x \sqrt{2\xi+1} f(\xi) d\xi \right.$$
$$\left. -\sqrt{2x+1} \int_x^N \sqrt{2\xi+1} f(\xi) d\xi \right) \tag{27}$$

$$\mathcal{A}^{\text{lowrank}}[f](x) = -\frac{1}{2}\sqrt{2x+1} \int_0^N \sqrt{2\xi+1} f(\xi) d\xi \tag{28}$$

Note that, as an operator, $\mathcal{A}^{\text{normal}}$ is normal, and $A^{\text{skew}}$ is skew-symmetric.

## 5. Analysis of Eigenvalues of HiPPO's Normal Part

Using the interpretation that CHIP Operator represents HiPPO when $N$ is large, this section explains that the conjecture on the normal part of the HiPPO matrix, Conjecture 2.5, can be resolved.

**Remark 5.1.** *CHIP Operator was defined as an operator on the space* $L^2([0, N])$, *but in this section we consider the limit* $N \to \infty$, *and therefore view it as an operator on* $L^2(\mathbb{R})$. *In particular, for* $f \in L^2([0, N])$, *we regard it as an element of* $L^2(\mathbb{R})$ *via the function* $f_{[0,N]}$ *supported on* $[0, N]$, *and note that* $L^2([0, N]) \subset L^2(\mathbb{R})$ *in this sense.*

### 5.1. Leading-Order Analysis

The conjecture concerns the imaginary parts of the eigenvalues $\lambda(A^{\text{normal}})$, but in fact it suffices to analyze the eigenvalues of $A^{\text{skew}}$ because the imaginary parts $\tau_n$ of the $\lambda(A^{\text{normal}})$ coincide with imaginary parts of $A^{\text{skew}}$. Here, instead of considering the skew-symmetric part $A^{\text{skew}}$ of the original HiPPO matrix, we consider the eigenvalues of the skew-symmetric part of CHIP Operator, namely those of $\mathcal{A}^{\text{skew}}$. In this case, the eigenvalues of $\mathcal{A}^{\text{skew}}$ can be written explicitly as follows.

**Proposition 5.2.** *The nonzero eigenvalues* $\lambda_k^* = i\tau_k^*$ $(k \in \mathbb{Z})$ *of* $\mathcal{A}^{\text{skew}}$ *are given as follows.*

$$\lambda_k^* = \frac{N^2 + N}{(2k+1)i\pi} \tag{29}$$

To compare the finite-dimensional HiPPO and the continuized HiPPO, we associate the finite-dimensional HiPPO matrix with the following operator.

$$\mathcal{A}_\Delta^{\text{skew}}[f](x) = -\int_0^N \sum_{i,j=0,\dots,N-1} \sqrt{i+\frac{1}{2}}\sqrt{j+\frac{1}{2}}$$
$$\text{sign}(i-j)\mathbb{1}_{[i,i+1)}(x)\mathbb{1}_{[j,j+1)}(\xi)f(\xi)d\xi \tag{30}$$

In particular, the eigenvalues of $\mathcal{A}_\Delta^{\text{skew}}$ and those of $A^{\text{skew}}$ are equal. At this point, one might hope that the operator norm of $\mathcal{A}_\Delta^{\text{skew}}$ and $\mathcal{A}^{\text{skew}}$ converges as $N \to \infty$, which would imply that their eigenvalues are asymptotically equal. However, for reasons such as a residual error remaining on each interval $[j, j+1)$, the difference in operator norm does not converge to zero. Therefore, instead of proving convergence of the norm of the difference, we show that it is bounded by $o(N^2)$.

**Proposition 5.3.**

$$\lim_{N\to\infty} \frac{1}{N^2} \left\| \mathcal{A}_\Delta^{\text{skew}} - \mathcal{A}^{\text{skew}} \right\|_{\mathscr{B}} = 0 \tag{31}$$

*That is,* $\left\| \mathcal{A}_\Delta^{\text{skew}} - \mathcal{A}^{\text{skew}} \right\|_{\mathscr{B}} = o(N^2)$

This $N^{-2}$ scaling is introduced because the eigenvalues in Proposition 5.2 grow on the order of $N^2$, so it rescales them so that they converge to constants in the eigenvalue scale. Indeed, this implies that the eigenvalues of the weighted HiPPO matrix $\frac{1}{N^2}A^{\text{skew}}$ are asymptotically equal to $\frac{1}{(2k+1)i\pi}$. In other words, the following theorem holds for the leading order.

**Theorem 5.4.** *Consider the eigenvalues of the $N \times N$ matrix $A^{\text{skew}}$, and let $\{\tau_n\}_{n=1...\lfloor \frac{N}{2} \rfloor}$ be the positive imaginary parts ordered monotonically decreasing in $n$. That is, $\tau_n > 0$ is the $n$-th largest among the positive imaginary parts of the eigenvalues. Then,*

$$\lim_{N \to \infty} \frac{\tau_n^{(N)}}{N^2} = \frac{1}{(2n-1)\pi} \tag{32}$$

*That is, as $N \to \infty$,*

$$\tau_n^{(N)} \sim \frac{N^2}{(2n-1)\pi} \tag{33}$$

Here, $\sim$ denotes that the leading asymptotic behavior agrees.

*Proof.* See Appendix E for the proof.

### 5.2. Sharp Asymptotic Analysis

Using the leading-order theorem above, one can also obtain the asymptotic behavior of lower-order terms.

**Theorem 5.5.** *Fixing each $n$, as $N \to \infty$,*

$$\tau_n^{(N)} = \frac{1}{(2n-1)\pi}N^2 - \frac{(2n-1)\pi}{6} + o(1) \tag{34}$$

*Proof.* See Appendix F for the proof.

This provides an answer to Conjecture 2.5. In particular, for the imaginary part of largest magnitude $\tau_1^{(N)}$,

$$\tau_1^{(N)} = \frac{N^2}{\pi} - \frac{\pi}{6} + o(1) \tag{35}$$

and $\frac{\pi}{6} \approx 0.5236$. For fixed $N$, the coefficient of the leading term $N^2$ is $\frac{1}{(2n-1)\pi}$, and one can also say that $\tau_n^{(N)} = \Theta(n^{-1})$. Note that the sign of the constant is reversed compared with conjecture. The proof proceeds by keeping the above integral interpretation in a discrete form and solving a differential equation (difference equation).

Let $\theta_n^{(N)}$ be defined by

$$\theta_n^{(N)} = \frac{N^2}{(2n-1)\pi} - \frac{(2n-1)\pi}{6}. \tag{36}$$

Figures 1, 2, 3, and 4 are plots of $\tau_n^{(N)} - \theta_n^{(N)}$ for $n = 1, 2, 4, 8$. These plots indicate that $\tau_n^{(N)} - \theta_n^{(N)} \to 0$ as $N \to \infty$ for all $n$.

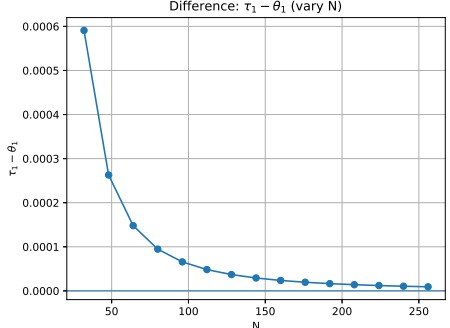

*Figure 1.* A plot of $\tau_1^{(N)} - \theta_1^{(N)}$ for increasing $N$

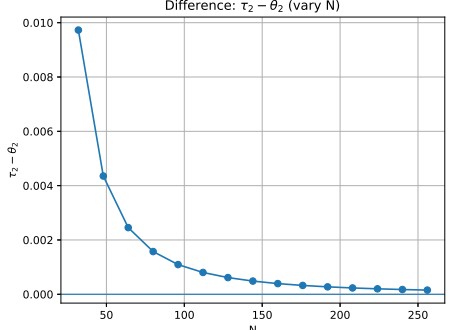

*Figure 2.* A plot of $\tau_2^{(N)} - \theta_2^{(N)}$ for increasing $N$

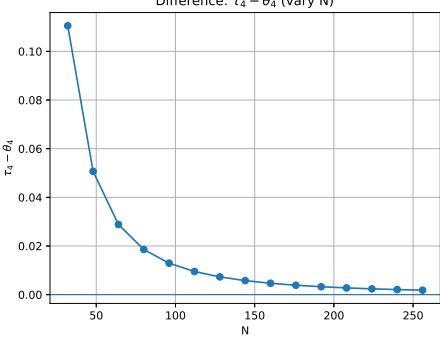

*Figure 3.* A plot of $\tau_4^{(N)} - \theta_4^{(N)}$ for increasing $N$

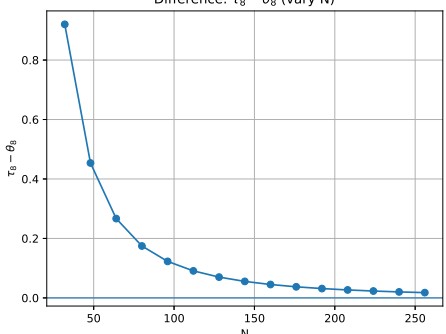

*Figure 4.* A plot of $\tau_8^{(N)} - \theta_8^{(N)}$ for increasing $N$

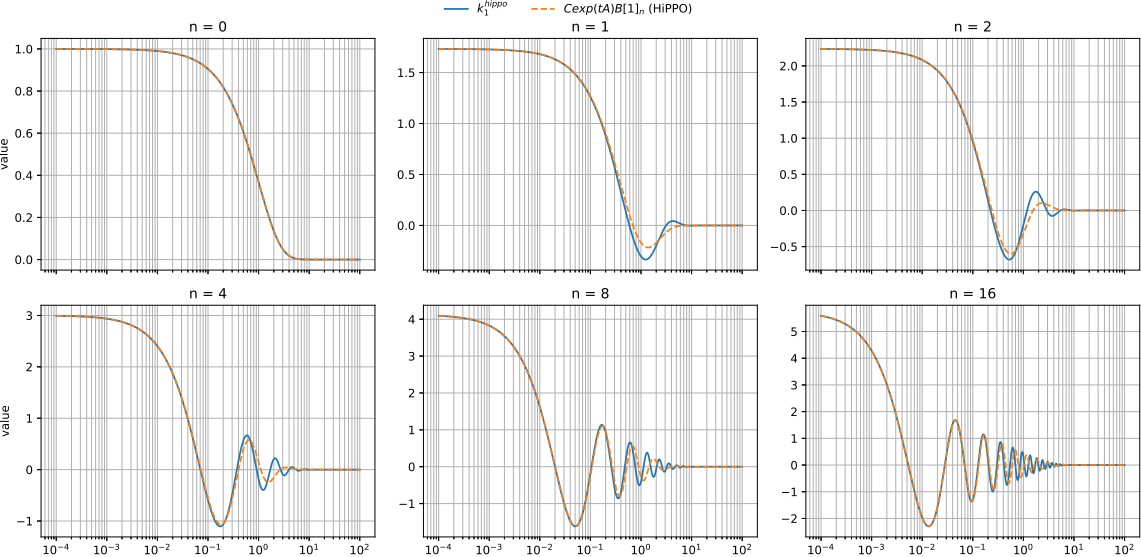

*Figure 5.* The comparison of $k_1^{\text{hippo}}$ with $c(x) = \delta_n(x)$ and $C\exp(A^{\text{hippo}})B[1]$ with $c_k = \delta_{kn}$.

## 6. An Explicit Representation of LSSL with Continuized-HiPPO Operator

In this section we show that the infinite-dimensional SSM using CHIP Operator admits an explicit integral representation.

### 6.1. Results

We first state the theorem. The LSSL using CHIP Operator above admits the following explicit representation of the solution.

**Theorem 6.1.** *If $v$ is Lipschitz continuous, then the strong solution of the LSSL using CHIP Operator is given by the following formula.*

$$y(t) = \int_0^t \int_0^N c(x) e^{-\frac{1}{2}(t-\tau)} \\ \sqrt{2x+1} J_0\left(2\sqrt{(t-\tau)}\sqrt{x^2+x}\right) v(\tau)\mathrm{d}x\mathrm{d}\tau \tag{37}$$

*Proof.* See Appendix G for the proof.

Here, $J_\nu$ is the Bessel function ([Olver et al., 2010](#)). This representation is written using only integrals, without using the operator exponential; in this sense, it is an explicit representation. Furthermore, this result can be interpreted as the SSM convolution kernel $k(t) = C\exp(tA)B$ being explicitly written as

$$k(t) = e^{-\frac{1}{2}t} \int_0^N c(x)\sqrt{2x+1} J_0\left(2\sqrt{t}\sqrt{x^2+x}\right) \mathrm{d}x. \tag{38}$$

### 6.2. State Space Model by Separable Volterra Operator

More generally, we give explicit solutions of SSMs for the family of operators defined below.

**Definition 6.2.** *Let $\mathcal{K}_{\varphi,\psi}$ be a separable Volterra integral operator. When an operator $\mathcal{F}_{\varphi,\psi,\omega}$ is given as follows*

$$\mathcal{F}_{\varphi,\psi,\omega} = \mathcal{K}_{\varphi,\psi} + \omega\mathcal{I} \tag{39}$$

*where $\varphi, \psi \in L^2, \omega \in \mathbb{R}$ and $\mathcal{K}_{\varphi,\psi}$ is defined in (18), we call it a regularized separable Volterra integral operator (RSV-IO).*

Note that CHIP Operator is an RSV-IO. Indeed, if we set $\chi(x) = \sqrt{2x+1}$, then

$$\mathcal{A}^{\text{hippo}}[f](x) = \left(\mathcal{K}_{i\chi,i\overline{\chi}} - \frac{1}{2}\mathcal{I}\right)[f](x) \tag{40}$$

which means $\mathcal{A}^{\text{hippo}} = \mathcal{K}_{i\chi,i\overline{\chi}} - \frac{1}{2}\mathcal{I}$. Here, $\overline{\chi}$ denotes the complex conjugate of $\chi$. Also, $\mathcal{B}^{\text{hippo}}$ can be expressed as the operator $\mathcal{G}_\chi[a](x) = a\chi(x)$. Hence, we can parameterize the operators using $\phi, \psi, \chi, \omega$ as parameters, which motivates the following definition.

**Definition 6.3.** *We call the pair consisting of an RSV-IO $\mathcal{F}_{\varphi,\psi,\omega}$ and the above $\mathcal{G}_\chi$ parametric CHIP.*

### 6.3. Explicit Solution by Parametric CHIP

LSSL by parametric CHIP:

$$\Phi(v) = C\tilde{h}, \ \tilde{h}(0)(x) = 0 \\ \frac{\partial \tilde{h}}{\partial t}(t) = \mathcal{F}_{\varphi,\psi,\omega}[\tilde{h}(t)] + \mathcal{G}_\chi[v(t)] \tag{41}$$

admits an explicit solution via the following explicit convolution kernel.

**Theorem 6.4.** *If $v$ is Lipschitz continuous, and $\mathcal{F}_{\varphi,\psi,\omega}$, and $\mathcal{G}_\chi$ are defined as above, then the SSM convolution kernel $k_{\varphi,\psi,\omega,\chi}$ of the LSSL using parametric CHIP $\mathcal{F}_{\varphi,\psi,\omega}, \mathcal{G}_\chi$ has the following explicit representation.*

$$k_{\varphi,\psi,\omega,\chi}(t) = e^{t\omega} \int_0^N c(x)\chi(x)\mathrm{d}x + e^{t\omega} \int_0^N \int_0^x$$

$$c(x)\varphi(x)\chi(\xi)\psi(\xi)\frac{tI_1\left(2\sqrt{t\int_\xi^x \varphi(s)\psi(s)\mathrm{d}s}\right)}{\sqrt{t\int_\xi^x \varphi(s)\psi(s)\mathrm{d}s}}\mathrm{d}\xi\mathrm{d}x,$$

$$(42)$$

*where, $I_\nu$ is the modified Bessel function (Olver et al., 2010).*

*Proof.* See Appendix G for the proof.

The essential reason this explicit solution can be obtained is the following lemma, which extends the repeated integration formula.

**Lemma 6.5.** *For $n \geq 1$, it holds that*

$$\mathcal{K}_{\varphi,\psi}^n [g](x)$$
$$= \frac{\varphi(x)}{(n-1)!} \int_0^x g(\xi)\psi(\xi) \left(\int_\xi^x \varphi(s)\psi(s)\mathrm{d}s\right)^{n-1} \mathrm{d}\xi.$$

$$(43)$$

The key point of this theorem is that a power of a separable Volterra Integral Operator is again a Volterra Integral Operator, and its kernel has a simple expression. Using this expression, we can compute $\exp(t\mathcal{K}_{\varphi,\psi})$ and obtain an integral representation of the SSM convolution kernel.

### 6.4. Explicit Solution by Continuized-HiPPO Operator

CHIP Operator has the form $\varphi = i\chi, \psi = i\overline{\chi}$. In this case, the kernel representation can be further simplified.

**Theorem 6.6.** *Assume $v$ is Lipschitz continuous. When $\varphi = i\chi, \psi = i\overline{\chi}$, the SSM convolution kernel $k_{i\chi,i\overline{\chi},\omega,\chi}$ of the LSSL using parametric CHIP $\mathcal{F}_{i\chi,i\overline{\chi},\omega}, \mathcal{G}_\chi$ has the following explicit representation.*

$$k_{i\chi,i\overline{\chi},\omega}(t) =$$
$$e^{\omega t} \int_0^N c(x)\chi(x)J_0\left(2\sqrt{t}\sqrt{\int_0^x |\chi(\xi)|^2 \mathrm{d}\xi}\right)\mathrm{d}x$$

$$(44)$$

*Proof.* See Appendix G for the proof.

By substituting $\chi(x) = \sqrt{2x+1}, \omega = -\frac{1}{2}$ into this expression, we obtain the explicit formula for the solution for CHIP Operator.

## 7. Comparison with the Original Kernel

We numerically compared the convolution kernels derived from the original HiPPO matrix with those from CHIP. When the input-function transformation

$$v(t) = -\frac{1}{2}\int_0^t e^{-(t-s)}u(s)ds + u(t) \qquad (45)$$

is applied, the explicit SSM convolution kernel $k_1^{\text{hippo}}$ can be written as

$$k_1^{\text{hippo}} = k_0^{\text{hippo}} * k_0 + k_0^{\text{hippo}} \qquad (46)$$

using $k_0^{\text{hippo}} = k_{i\chi,i\overline{\chi},\omega}$ and $k_0(t) = -\frac{1}{2}e^{-t}$. We compare the kernel $k_1^{\text{hippo}}$ obtained by setting the kernel parameter $c$ to $c(x) = \delta_n(x)$ with the original kernel $C\exp(A^{\text{hippo}})B[1]$ obtained by setting the parameter vector to $c_k = \delta_{kn}$. Note that since the explicit kernel was expressed as the inner product of a continuous function and the parameter $c$, $c$ can be the Dirac delta function, according to the proofs of theorems. The comparison in a logarithmic time-scale is shown in Figure 5 at $n = 0, 1, 2, 4, 8, 16$. For details, see Appendix B.

In Figure 5, although the phase of the wave is shifted, the scaling remains unchanged; this can be interpreted as follows. The regularization term $\frac{t}{2}I$ in HiPPO affects the integral kernel through the factor $\exp\left(-\frac{t}{2}I\right)$. On the other hand, the skew-symmetric component $A^{\text{skew}}$ affects it through $\exp\left(tA^{\text{skew}}\right)$. Suppose that the low-rank part $A^{\text{lowrank}}$ can be neglected as $N \to \infty$. Note that the eigenvalues of $\frac{t}{2}I$ are real, whereas those of $A^{\text{skew}}$ are purely imaginary. $\exp\left(tA^{\text{skew}}\right)$ corresponds to $\exp\left(t\mathcal{A}^{\text{skew}}\right)$ in the continuous setting, and the discrepancy induced by continuousization appears as the exponential of a purely imaginary quantity, that is, as a phase shift. On the other hand, $\exp\left(-\frac{t}{2}I\right)$ corresponds, in the continuous setting, to $\exp\left(-\frac{t}{2}\mathcal{I}\right)$ and is therefore not influenced by the discretization-to-continuum transition.

## 8. Conclusion

In this paper, we proposed the CHIP Operator that associates the HiPPO Matrix with an operator on the function space $L^2$. In particular, using this idea we analyzed the asymptotic behavior of eigenvalues of the normal part of HiPPO. Moreover, we showed that the LSSL induced by CHIP Operator admits an explicit representation of the solution. For future work, the results on asymptotic analysis may also be applicable to state space models for other matrices. In addition, the explicit-solution result provides a form that is convenient for model construction and theoretical analysis, and may lead to new implementations or interpretations.

State Space Model with Continuous Limit of HiPPO Matrix: Eigenvalue Analysis and Explicit Solution Formula

## Impact Statement

This paper presents work whose goal is to advance the field of Machine Learning. There are many potential societal consequences of our work, none which we feel must be specifically highlighted here.

## Acknowledgement

Funding in direct support of this work: JST CREST Grant Number JPMJCR24Q5, JSPS KAKENHI Grant Number 25K15148, JST ASPIRE JPMJAP2329. This work is also supported by the Horizon Europe, MSCA-SE project 101131557 (REMODEL.)

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

# A. A Supplementary Experiment

This experiment is intended as a sanity check demonstrating that the explicit-kernel formulation can be incorporated into a standard learning pipeline.

We present a numerical experiment on Sequential MNIST, comparing a model based on the explicit solution with an existing HiPPO-based S4 model.

## A.1. A Layer Based on the Explicit Kernel

As a layer derived from the explicit solution, we used the following architecture. The kernel derived from the explicit representation

$$e^{-\frac{1}{2}t} \int_0^N c(x)\sqrt{2x+1}J_0\left(2\sqrt{t}\sqrt{x^2+x}\right) \mathrm{d}x \tag{47}$$

is discretized with $\Delta x = 1$ as follows.

$$k_l = \Delta t e^{-\frac{1}{2}l\Delta t} \sum_{n=0}^{N-1} c_n \sqrt{2n\Delta x + 1} J_0\left(2\sqrt{l\Delta t}\sqrt{(n\Delta x)^2 + n\Delta x}\right)\Delta x \tag{48}$$

$$= \Delta t e^{-\frac{1}{2}l\Delta t} \sum_{n=0}^{N-1} c_n \sqrt{2n+1} J_0\left(2\sqrt{l\Delta t}\sqrt{n^2 + n}\right) \tag{49}$$

Note that the discretization width $\Delta t$ arising from the discretization of the continuous-time convolution is included in this discrete kernel. We compared the original S4 model with a model in which the SSM convolution kernel of the S4 model is replaced by this kernel $k = (k_l)_{l=0}^{L-1}$. The vector $c = (c_k)_{k=0}^{N-1}$ and $\Delta t$ are trainable parameters. In addition, in the architecture with the explicit kernel, we multiplied the Fourier transform of the convolution kernel by

$$V_{l'} = \mathrm{FFT}\left(\mathrm{Padding}\left(\left(-\frac{1}{2}e^{-k\Delta t}\Delta t\right)_{k=0}^{L-1}\right)\right)_{l'} + 1. \tag{50}$$

This reflects the replacement of the input function

$$v(t) = -\frac{1}{2}\int_0^t e^{-(t-s)}u(s)ds + u(t). \tag{51}$$

That is, given the convolution kernel $K_l$, the output is computed as in Algorithm 1.

## A.2. An Experiment and Algorithms

Since the explicit-kernel model is derived from the fixed Continuized-HiPPO operator and does not learn the state matrix A or input matrix B directly, we compare it with two S4-style baselines: one with fixed HiPPO A and B, and one in which A and B are learnable. Algorithms 1, 2, and 3 show how the layer outputs are computed.

---

**Algorithm 1** SSM by Explicit Kernel

---

   **Input:** $u \in \mathbb{R}^L$
   **Trainable Parameters:** $c \in \mathbb{R}^N, \Delta t \in \mathbb{R}$
   **Initialize:** $c_n \sim \mathcal{N}(0,1), \log(\Delta t) \sim \mathcal{U}(\log(0.001), \log(0.1))$
   Compute $k = (k_l)_{l=0,\ldots,L-1}$
   $K_{l'} \leftarrow \mathrm{FFT}(\mathrm{Padding}(k))_{l'}$
   $U_{l'} \leftarrow \mathrm{FFT}(\mathrm{Padding}(u))_{l'}$
   Compute $V_{l'}$
   $Y_{l'} \leftarrow U_{l'}V_{l'}K_{l'}$
   $y_l \leftarrow \mathrm{IFFT}(Y)_l$
   **Return:** $y \in \mathbb{R}^L$

---

**Algorithm 2** S4 Model (Fixed)

---

**Input:** $u$
**Trainable Parameters:** $c \in \mathbb{R}^N, \Delta t \in \mathbb{R}$
**Initialize:** $c_n \sim \mathcal{N}(0, 1), \log(\Delta t) \sim \mathcal{U}(\log(0.001), \log(0.1))$
**Fixed Parameters:** $A, B$: HiPPO matrices
Compute $\tilde{k} = \left(\tilde{k}_l(A, B)\right)_{l=0,\ldots,L-1}$ using the S4 model (depends on $\Delta t, c$)
$K_{l'} \leftarrow \text{FFT}\left(\text{Padding}(\tilde{k})\right)_{l'}$
$U_{l'} \leftarrow \text{FFT}\left(\text{Padding}(u)\right)_{l'}$
$Y_{l'} \leftarrow U_{l'} K_{l'}$
$y_l \leftarrow \text{IFFT}(Y)_l$
**Return:** $y$

---

**Algorithm 3** S4 Model (Learnable)

---

**Input:** $u$
**Trainable Parameters:** $c \in \mathbb{R}^N, \Delta t \in \mathbb{R}$, DPLR parameters for A, and $B \in \mathbb{R}^N$
**Initialize:** $c_n \sim \mathcal{N}(0, 1), \log(\Delta t) \sim \mathcal{U}(\log(0.001), \log(0.1)), (A, B)$: HiPPO matrices
Compute $\tilde{k} = \left(\tilde{k}_l(A, B)\right)_{l=0,\ldots,L-1}$ using the S4 model (depends on $\Delta t, c$)
$K_{l'} \leftarrow \text{FFT}\left(\text{Padding}(\tilde{k})\right)_{l'}$
$U_{l'} \leftarrow \text{FFT}\left(\text{Padding}(u)\right)_{l'}$
$Y_{l'} \leftarrow U_{l'} K_{l'}$
$y_l \leftarrow \text{IFFT}(Y)_l$
**Return:** $y$

---

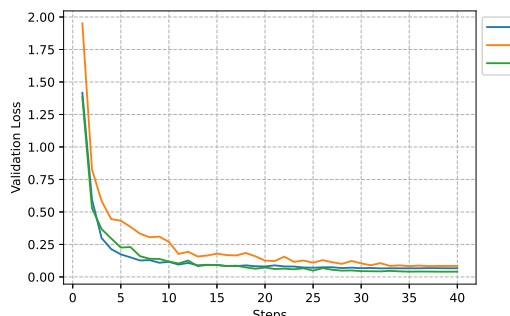

*Table 1.* Final validation accuracy and validation loss.

|  | Accuracy | Validation loss |
|---|---|---|
| S4 model (learnable) | 0.987 | 0.0404 |
| S4 model (fixed) | 0.9732 | 0.0847 |
| Explicit-kernel model | 0.9785 | 0.0666 |

*Figure 6.* Validation loss curves during training on Sequential MNIST.

## A.3. Details

All three models share the same overall architecture. The model consists of an input layer, an intermediate sequence layer, and an output layer. The only difference among the three models is how the convolution kernel in the intermediate layer is computed. The input layer was implemented as a linear layer, while the output layer consists of a temporal averaging layer followed by a linear layer. The intermediate layer was an S4-style sequence layer with a skip connection. In addition, a batch normalization layer was applied before the S4 layer, the activation function of the S4 output was the Gaussian Error Linear Unit (GELU)(Hendrycks & Gimpel, 2016), and after the skip connection, a Gated Linear Unit (GLU)(Dauphin et al., 2017) was used. We set the model dimension to $d_{model} = 64$ and the dimension of the state space to $N = 64$, and initialized the parameters as follows: $c_n \sim \mathcal{N}(0, 1), \log(\Delta t) \sim \mathcal{U}(\log(0.001), \log(0.1))$. We used the Adam optimizer (Kingma & Ba, 2015), together with a cosine annealing scheduler with warmup. The learning rate was 0.001. The number of warmup steps was set to 1200. The batch size was 100 and the number of epochs was 40. We used the torchvision MNIST dataset and flattened each 28 !_28 image into a length-784 sequence. For the experimental

environment, we used PyTorch as the library, and conducted training on an NVIDIA GeForce RTX 3060 Laptop GPU.

### A.4. Results

We show the validation loss curves in Figure 6 and the final validation accuracy and loss in Table 1. This suggests that, at least under the above settings, the results are comparable to those obtained using the S4 model with fixed $A$ and $B$.

## B. About Figure 5

We calculated $k_0^{\text{hippo}} * k_0$ as follows. By defining $h$ as

$$h(t) = k_0^{\text{hippo}} * k_0 = -\frac{1}{2} \int_0^t e^{-(t-s)} k_0^{\text{hippo}}(s) \mathrm{d}s \tag{52}$$

and differentiating both sides by $t$, it holds that

$$\begin{aligned}
\frac{\partial}{\partial t} h(t) &= -\frac{1}{2} e^{-(t-t)} k_0^{\text{hippo}}(t) + \frac{1}{2} \int_0^t e^{-(t-s)} k_0^{\text{hippo}}(s) \mathrm{d}s \\
&= -\frac{1}{2} k_0^{\text{hippo}}(t) - h(t).
\end{aligned} \tag{53}$$

This is an ordinary differential equation for $h$, and we plotted $k_1^{\text{hippo}}(t) = h(t) + k_0^{\text{hippo}}$ discretized using RK4 method.

## C. The Proof of Lemma 2.2

The following is known as a standard result in semigroup theory.

**Theorem C.1** ((Pazy, 1983), section 4-2, Cor 2.11)**.** *Let $A$ be the generator of a semigroup $\{T(t)\}_{t \geq 0}$ on a reflexive Banach space $X$, and let $h_0 \in \mathcal{D}(A)$ be the initial value. If $f$ is Lipschitz continuous on $[0,T]$, then*

$$\begin{aligned}
\frac{dh}{dt} &= Ah + f \\
h(0) &= h_0
\end{aligned} \tag{54}$$

*the strong solution is given on $t \in [0,T]$ by*

$$h(t) = T(t)h_0 + \int_0^t T(t-s)f(s)ds$$

*uniquely.*

Using this, Lemma 2.2 can be proved.

*Proof of Lemma 2.2.* A Hilbert space is a reflexive Banach space. Moreover, if $A$ is a bounded linear operator, then $\exp(tA)$ can be defined as a bounded linear operator; hence, $\{\exp(tA)\}_{t \geq 0}$ forms a semigroup with generator $A$. Also, the initial value $0 \in \mathcal{D}(A)$ holds. Therefore, it suffices to verify the Lipschitz continuity of $Bu$. Noting that $B$ is bounded, i.e., $\|B\| < \infty$, and that $u$ is assumed Lipschitz continuous,

$$\begin{aligned}
\|Bu(t) - Bu(s)\| &\leq \|B\| \, \|u(t) - u(s)\| \\
&\leq L\|B\||t - s|
\end{aligned} \tag{55}$$

holds for any $s, t$. That is, $Bu$ is Lipschitz continuous. $\qquad\square$

## D. The Proof of Theorem 4.1

*Proof.* Writing (20) componentwise, for $i > 0$ we obtain the following by algebraic manipulation.

$$
\begin{aligned}
\frac{\partial h_i}{\partial t} =& -\sqrt{2i+1}h_0 - \sqrt{2i+1}\sum_{l=1}^{i-1}\sqrt{2l+1}h_l - (i+1)h_i + \sqrt{2i+1}u(t) \\
=& -\sqrt{2i+1}\cdot\frac{h_0}{2} - \sqrt{2i+1}\sum_{l=1}^{i-1}\sqrt{2l+1}h_l - \left(i+\frac{1}{2}\right)h_i - \frac{h_i}{2} \\
& -\sqrt{2i+1}\cdot\frac{h_0}{2} + \sqrt{2i+1}u \\
=& -\sqrt{2i+1}\cdot\frac{h_0}{2} - \sqrt{2i+1}\sum_{l=1}^{i-1}\sqrt{2l+1}h_l - (2i+1)\frac{h_i}{2} - \frac{h_i}{2} \\
& -\sqrt{2i+1}\cdot\frac{h_0}{2} + \sqrt{2i+1}u \\
=& -\sqrt{2i+1}\left(\frac{h_0}{2} + \sum_{l=1}^{i-1}\sqrt{2l+1}h_l + \sqrt{2i+1}\frac{h_i}{2}\right) - \frac{h_i}{2} \\
& -\sqrt{2i+1}\cdot\frac{h_0}{2} + \sqrt{2i+1}u
\end{aligned}
\tag{56}
$$

Here, when $i = 0$, since

$$
\frac{\partial h_0}{\partial t} = -h_0 + u(t)
\tag{57}
$$

holds, by solving it, which can be written as

$$
h_0(t) = \int_0^t e^{-(t-s)}u(s)\mathrm{d}s.
\tag{58}
$$

Therefore, substituting this into (56) yields

$$
\begin{aligned}
\frac{\partial h_i}{\partial t}(t) =& -\sqrt{2i+1}\left(\frac{h_0(t)}{2} + \sum_{l=1}^{i-1}\sqrt{2l+1}h_l(t) + \sqrt{2i+1}\frac{h_i(t)}{2}\right) - \frac{h_i(t)}{2} \\
& -\sqrt{2i+1}\cdot\frac{h_0(t)}{2} + \sqrt{2i+1}u(t) \\
=& -\sqrt{2i+1}\left(\frac{h_0(t)}{2} + \sum_{l=1}^{i-1}\sqrt{2l+1}h_l(t) + \sqrt{2i+1}\frac{h_i(t)}{2}\right) - \frac{h_i(t)}{2} \\
& -\sqrt{2i+1}\cdot\frac{1}{2}\int_0^t e^{-(t-s)}u(s)ds + \sqrt{2i+1}u(t) \\
=& -\sqrt{2i+1}\left(\frac{h_0(t)}{2} + \sum_{l=1}^{i-1}\sqrt{2l+1}h_l(t) + \sqrt{2i+1}\frac{h_i(t)}{2}\right) - \frac{h_i(t)}{2} \\
& +\sqrt{2i+1}\left(-\frac{1}{2}\int_0^t e^{-(t-s)}u(s)ds + u(t)\right)
\end{aligned}
\tag{59}
$$

Especially, first term can be viewed as a discretization of the numerical integration in (21) by the trapezoidal rule. □

## E. Proof of Theorem 5.4

To begin with, we describe the relationship between the matrix $A^{\text{skew}}$ and the operator $\mathcal{A}_\Delta^{\text{skew}}$.

The skew-symmetric part of HiPPO $A^{\text{skew}}$ is given by

$$
A_{ij}^{\text{skew}} = -\frac{1}{2}\sqrt{2i+1}\sqrt{2j+1}\,\text{sign}(i-j) = -\sqrt{i+\frac{1}{2}}\sqrt{j+\frac{1}{2}}\,\text{sign}(i-j) \qquad (i, j = 0, \ldots, N-1)
\tag{60}
$$

and $A_\Delta^{\text{skew}}$ is given by

$$\mathcal{A}_\Delta^{\text{skew}}[f](x) = -\int_0^N \sum_{i,j=0,\ldots,N-1} \sqrt{i+\frac{1}{2}}\sqrt{j+\frac{1}{2}} \operatorname{sign}(i-j)\mathbb{1}_{[i,i+1)}(x)\mathbb{1}_{[j,j+1)}(\xi)f(\xi)\mathrm{d}\xi. \tag{61}$$

Let $I_i = [i, i+1)$, and define the $N$-dimensional subspace $V_N \subset L^2(\mathbb{R})$ by

$$V_N := \left\{ \sum_{j=0}^{N-1} c_j \mathbb{1}_{I_j} \;\middle|\; (c_0,\ldots,c_{N-1}) \in \mathbb{C}^N \right\} \subset L^2(\mathbb{R}) \tag{62}$$

Then the operator $\mathcal{A}_\Delta^{\text{skew}}$ is $V_N$ invariant and the matrix representation of $\mathcal{A}_\Delta^{\text{skew}}\big|_{V_N}$ on $V_N$ is equal to $A^{\text{skew}}$. In particular, $\mathcal{A}_\Delta^{\text{skew}}$, $\mathcal{A}_\Delta^{\text{skew}}\big|_{V_N}$, and $A^{\text{skew}}$ have the same eigenvalues, described by next lemma.

**Lemma E.1.** *The operator obtained by restricting $\mathcal{A}^{\text{skew}}$ to $V_N$, namely $\mathcal{A}_\Delta^{\text{skew}}\big|_{V_N}$, has representation matrix with respect to the basis $\{\mathbb{1}_{I_j}\}_{j=0}^{N-1}$ that coincides with $A^{\text{skew}}$. In particular, all nonzero eigenvalues of $\mathcal{A}_\Delta^{\text{skew}}$ and $A^{\text{skew}}$ coincide.*

*Proof.* For each basis element $\mathbb{1}_{I_i}$ $(j = 0, ..., N-1)$ of $V_N$, since

$$\begin{aligned}
\mathcal{A}_\Delta^{\text{skew}}\big|_{V_N}[\mathbb{1}_{I_i}] &= -\sum_{j=0}^{N-1} \sqrt{i+\frac{1}{2}}\sqrt{j+\frac{1}{2}} \operatorname{sign}(i-j)\mathbb{1}_{I_i}(x)\int_0^N \mathbb{1}_{I_i}(\xi)\mathrm{d}\xi \\
&= -\sum_{j=0}^{N-1} \sqrt{i+\frac{1}{2}}\sqrt{j+\frac{1}{2}} \operatorname{sign}(i-j)\mathbb{1}_{I_i}(x)
\end{aligned} \tag{63}$$

holds, the $\mathbb{1}_{I_i}$'s cofficients are written as

$$-\sqrt{i+\frac{1}{2}}\sqrt{j+\frac{1}{2}} \operatorname{sign}(i-j) \tag{64}$$

This matches the componentwise expression of $A^{\text{skew}}$.

Next, we consider eigenvalues. From the above, since the representation matrices coincide, the eigenvalues of $\mathcal{A}_\Delta^{\text{skew}}$ coincide with the nonzero eigenvalues of $\mathcal{A}_\Delta^{\text{skew}}\big|_{V_N}$. On the other hand, by definition, for any $f \in L^2(\mathbb{R})$, $\mathcal{A}_\Delta^{\text{skew}}[f] \in V_N$. Hence, for any nonzero eigenvalue $\lambda \neq 0$ and corresponding eigenvector $f \in L^2(\mathbb{R})$, we have $f = \lambda^{-1}\mathcal{A}_\Delta^{\text{skew}}[f] \in V_N$. Therefore, every eigenvalue of $\mathcal{A}_\Delta^{\text{skew}}$ is an eigenvalue of $\mathcal{A}_\Delta^{\text{skew}}\big|_{V_N}$. Conversely, it is clear that any eigenvalue of $\mathcal{A}_\Delta^{\text{skew}}\big|_{V_N}$ is also an eigenvalue of $\mathcal{A}_\Delta^{\text{skew}}$; hence, the eigenvalues of $\mathcal{A}_\Delta^{\text{skew}}\big|_{V_N}$ and those of $\mathcal{A}_\Delta^{\text{skew}}$ coincide. $\square$

From here on, we analyze asymptotic behavior of the difference between the continuous operator $\mathcal{A}^{\text{skew}} \in \mathbb{R}^{N \times N}$ and the discrete operator $\mathcal{A}_\Delta^{\text{skew}} \in \mathbb{R}^{N \times N}$, and of the difference of their eigenvalues; for convenience, we consider the following scaled operators.

$$\mathcal{S}^{(N)} := \frac{1}{N^2+N}\mathcal{A}^{\text{skew}}, \quad \mathcal{S}_\Delta^{(N)} := \frac{1}{N^2+N}\mathcal{A}_\Delta^{\text{skew}} \tag{65}$$

Note that the eigenvalues of each operator are also scaled by $(N^2+N)^{-1}$. The following proposition states that the difference of the scaled operators converges to 0 as $N \to \infty$.

**Proposition E.2.** *In terms of convergence in operator norm,*

$$\lim_{N \to \infty} \left\| \mathcal{S}_\Delta^{(N)} - \mathcal{S}^{(N)} \right\|_{\mathscr{B}(L^2(\mathbb{R}),L^2(\mathbb{R}))} = 0 \tag{66}$$

*holds.*

*Proof.* In what follows, define $\chi_0$ as $\chi_0(x) := \sqrt{x + \frac{1}{2}}$. By definition, $\mathcal{S}^{(N)}$ and $\mathcal{S}_\Delta^{(N)}$ can be written as integral operators with the following kernels $K^{(N)}$ and $K_\Delta^{(N)}$, respectively.

$$
\begin{aligned}
K^{(N)}(x, \xi) &= -\frac{1}{N^2 + N} \sqrt{x + \frac{1}{2}} \sqrt{\xi + \frac{1}{2}} \operatorname{sign}(x - \xi) \mathbb{1}_{[0,N)}(x) \mathbb{1}_{[0,N)}(\xi) \\
&= -\frac{1}{N^2 + N} \chi_0(x) \chi_0(\xi) \operatorname{sign}(x - \xi) \mathbb{1}_{[0,N)}(x) \mathbb{1}_{[0,N)}(\xi)
\end{aligned}
\tag{67}
$$

$$
\begin{aligned}
K_\Delta^{(N)}(x, \xi) &= -\frac{1}{N^2 + N} \sum_{i,j=0,\ldots,N-1} \sqrt{i + \frac{1}{2}} \sqrt{j + \frac{1}{2}} \operatorname{sign}(i - j) \mathbb{1}_{[i,i+1)}(x) \mathbb{1}_{[j,j+1)}(\xi) \\
&= -\frac{1}{N^2 + N} \sum_{i,j=0,\ldots,N-1} \chi_0(i) \chi_0(j) \operatorname{sign}(i - j) \mathbb{1}_{[i,i+1)}(x) \mathbb{1}_{[j,j+1)}(\xi)
\end{aligned}
\tag{68}
$$

Moreover, the operator $\mathcal{S}^{(N)} - \mathcal{S}_\Delta^{(N)}$ is also an integral operator with kernel $K^{(N)} - K_\Delta^{(N)}$. so it is an integral operator induced by this kernel. Here we evaluate the Hilbert–Schmidt (HS) norm of $\mathcal{S}^{(N)} - \mathcal{S}_\Delta^{(N)}$, i.e., the kernel $K^{(N)} - K_\Delta^{(N)}$'s $L^2(\mathbb{R}^2)$ norm, and estimate it. If we can show that this norm converges to 0, then the following bound implies that the operator norm also converges to 0.

$$
\left\| \mathcal{S}^{(N)} - \mathcal{S}_\Delta^{(N)} \right\|_{\mathcal{B}(L^2(\mathbb{R}), L^2(\mathbb{R}))} \leq \left\| \mathcal{S}^{(N)} - \mathcal{S}_\Delta^{(N)} \right\|_{HS} = \left\| K^{(N)} - K_\Delta^{(N)} \right\|_{L^2(\mathbb{R}^2)}
\tag{69}
$$

Therefore, it suffices to show that the following integral converges to 0; this establishes (66).

$$
\left\| K^{(N)} - K_\Delta^{(N)} \right\|_{L^2(\mathbb{R}^2)}^2 = \int_0^N \int_0^N \left| K^{(N)}(x, \xi) - K_\Delta^{(N)}(x, \xi) \right|^2 \mathrm{d}\xi \mathrm{d}x
\tag{70}
$$

Accordingly, in what follows we show that

$$
\lim_{N \to \infty} \int_0^N \int_0^N \left| K^{(N)}(x, \xi) - K_\Delta^{(N)}(x, \xi) \right|^2 \mathrm{d}\xi \mathrm{d}x = 0
\tag{71}
$$

(71). We split the integration domain $[0, N)^2 = \bigsqcup_{i,j=0,\ldots,N-1} [i, i+1) \times [j, j+1)$ and analyze on each piece. In doing so, we use the following two function estimates. First, for any $i \in \mathbb{Z}_{\geq 0}$ and $x \in [i, i+1)$, $\chi_0'(x) = \frac{1}{2\sqrt{x + \frac{1}{2}}} \leq \frac{1}{\sqrt{2}}$ Using this, the mean value theorem implies that there exists $\theta \in (x, i)$ such that the following holds.

$$
\begin{aligned}
|\chi_0(x) - \chi_0(i)| &= |\chi_0'(\theta)(x - i)| \\
&\leq |\chi_0'(\theta)| |x - i| \\
&\leq \frac{1}{2\sqrt{x + \frac{1}{2}}} \cdot 1 \\
&\leq \frac{1}{\sqrt{2}}
\end{aligned}
\tag{72}
$$

Second, if $N \geq \frac{1}{2}$, then for any $x \in [0, N)$, $\chi_0(x) = \sqrt{x + \frac{1}{2}} \leq \sqrt{N + \frac{1}{2}} \leq \sqrt{2N}$.

Below we analyze each integral piece.

1. When $i \neq j$, on $(x, \xi) \in [i, i+1) \times [j, j+1)$, since $\operatorname{sign}(x - \xi) = \operatorname{sign}(i - j)$, we obtain

$$
K^{(N)}(x, \xi) - K_\Delta^{(N)}(x, \xi) = -\frac{\operatorname{sign}(i - j)}{N^2 + N} \Big( \chi_0(x) \chi_0(\xi) - \chi_0(i) \chi_0(j) \Big).
\tag{73}
$$

In this case,

$$
\begin{aligned}
&|\chi_0(x)\chi_0(\xi) - \chi_0(i)\chi_0(j)| \\
&= |\chi_0(x)\chi_0(\xi) - \chi_0(i)\chi_0(\xi) + \chi_0(i)\chi_0(\xi) - \chi_0(i)\chi_0(j)| \\
&\le |\chi_0(x) - \chi_0(i)|\,|\chi_0(\xi)| + |\chi_0(i)|\,|\chi_0(\xi) - \chi_0(j)| \\
&\le \frac{1}{\sqrt{2}}\sqrt{2N} + \sqrt{2N}\frac{1}{\sqrt{2}} \\
&\le 2\sqrt{N}.
\end{aligned}
\tag{74}
$$

Therefore the $L^2\big([i, i+1) \times [j, j+1)\big)$ norm becomes as follows.

$$
\begin{aligned}
&\int_i^{i+1}\int_j^{j+1} \left|K^{(N)}(x,\xi) - K_\Delta^{(N)}(x,\xi)\right|^2 \mathrm{d}\xi\mathrm{d}x \\
&= \int_i^{i+1}\int_j^{j+1} \left|\frac{\operatorname{sign}(i-j)}{N^2+N}\Big(\chi_0(x)\chi_0(\xi) - \chi_0(i)\chi_0(j)\Big)\right|^2 \mathrm{d}\xi\mathrm{d}x \\
&= \frac{1}{(N^2+N)^2}\int_i^{i+1}\int_j^{j+1} |\chi_0(x)\chi_0(\xi) - \chi_0(i)\chi_0(j)|^2 \,\mathrm{d}\xi\mathrm{d}x \\
&\le \frac{1}{(N^2+N)^2}\int_i^{i+1}\int_j^{j+1} \left(2\sqrt{N}\right)^2 \mathrm{d}\xi\mathrm{d}x \\
&= \frac{4N}{(N^2+N)^2}
\end{aligned}
\tag{75}
$$

2. When $i = j$, on $(x, \xi) \in [i, i+1) \times [j, j+1) = [i, i+1)^2$, $\operatorname{sign}(i-j) = \operatorname{sign}(i-i) = 0$ holds, and so

$$
K^{(N)}(x,\xi) - K_\Delta^{(N)}(x,\xi) = K^{(N)}(x,\xi) - 0 = -\frac{\operatorname{sign}(x-\xi)}{N^2+N}\chi_0(x)\chi_0(\xi)
\tag{76}
$$

holds. Therefore the $L^2\big([i, i+1) \times [j, j+1)\big)$ norm becomes as follows.

$$
\begin{aligned}
&\int_i^{i+1}\int_j^{j+1} \left|K^{(N)}(x,\xi) - K_\Delta^{(N)}(x,\xi)\right|^2 \mathrm{d}\xi\mathrm{d}x \\
&= \int_i^{i+1}\int_j^{j+1} \left|\frac{\operatorname{sign}(x-\xi)}{N^2+N}\chi_0(x)\chi_0(\xi)\right|^2 \mathrm{d}\xi\mathrm{d}x \\
&= \frac{1}{(N^2+N)^2}\int_i^{i+1}\int_j^{j+1} |\chi_0(x)\chi_0(\xi)|^2 \,\mathrm{d}\xi\mathrm{d}x \\
&\le \frac{1}{(N^2+N)^2}\int_i^{i+1}\int_j^{j+1} \left(\sqrt{2N}\cdot\sqrt{2N}\right)^2 \mathrm{d}\xi\mathrm{d}x \\
&= \frac{4N^2}{(N^2+N)^2}
\end{aligned}
\tag{77}
$$

Using these, we can estimate the integral (70) as follows.

$$
\begin{aligned}
0 &\leq \int_0^N \int_0^N \left| K^{(N)}(x,\xi) - K_\Delta^{(N)}(x,\xi) \right|^2 \mathrm{d}\xi \mathrm{d}x \\
&= \sum_{\substack{i,j=0,\ldots,N-1 \\ i \neq j}} \int_i^{i+1} \int_j^{j+1} \left| K^{(N)}(x,\xi) - K_\Delta^{(N)}(x,\xi) \right|^2 \mathrm{d}\xi \mathrm{d}x \\
&\quad + \sum_{i=0}^{N-1} \int_i^{i+1} \int_i^{i+1} \left| K^{(N)}(x,\xi) - K_\Delta^{(N)}(x,\xi) \right|^2 \mathrm{d}\xi \mathrm{d}x \\
&\leq \frac{4N}{(N^2+N)^2} \sum_{\substack{i,j=0,\ldots,N-1 \\ i \neq j}} 1 + \frac{4N^2}{(N^2+N)^2} \sum_{i=0}^{N-1} 1 \\
&= \frac{4N \cdot (N^2-N) + 4N^2 \cdot N}{(N^2+N)^2} \xrightarrow{N\to\infty} 0
\end{aligned}
\tag{78}
$$

Especially, (71) holds. $\qquad\square$

Note that Proposition 5.3 is a rephrasing of this Proposition E.2.

*Proof of Proposition 5.3.* Using Proposition E.2, the following holds.

$$
\begin{aligned}
\lim_{N\to\infty} \frac{1}{N^2} \left\| \mathcal{A}_\Delta^{\mathrm{skew}} - \mathcal{A}^{\mathrm{skew}} \right\|_{\mathscr{B}} &= \lim_{N\to\infty} \frac{N^2+N}{N^2} \left\| \frac{1}{N^2+N} \mathcal{A}_\Delta^{\mathrm{skew}} - \frac{1}{N^2+N} \mathcal{A}^{\mathrm{skew}} \right\|_{\mathscr{B}} \\
&= \lim_{N\to\infty} \frac{N^2+N}{N^2} \left\| \mathcal{S}_\Delta^{\mathrm{skew}} - \mathcal{S}^{\mathrm{skew}} \right\|_{\mathscr{B}} \\
&= 1 \cdot 0 \\
&= 0
\end{aligned}
\tag{79}
$$

$\qquad\square$

The next proposition states that the eigenvalues of the scaled continuous operator $\mathcal{S}^{(N)}$ can be written explicitly.

**Proposition E.3.** *The nonzero eigenvalues $\{\lambda_k^{\mathcal{S}^{(N)}}\}_{k\in\mathbb{Z}}$ of the operator $\mathcal{S}^{(N)}$ are given by*

$$
\lambda_k^{\mathcal{S}^{(N)}} = -\frac{1}{(2k+1)i\pi} \qquad (k \in \mathbb{Z})
\tag{80}
$$

*Proof.* Assume that the eigenvalue equation $\mathcal{S}^{(N)}[f] = \lambda f$ holds. That is, for $x \in [0, N)$,

$$
\frac{1}{N^2+N} \left( -\sqrt{x+\frac{1}{2}} \int_0^x \sqrt{\xi+\frac{1}{2}} f(\xi) \mathrm{d}\xi + \sqrt{x+\frac{1}{2}} \int_x^N \sqrt{\xi+\frac{1}{2}} f(\xi) \mathrm{d}\xi \right) = \lambda f(x)
\tag{81}
$$

is satisfied. When $\lambda \neq 0$, each integral term on the left-hand side is continuous for any $x \in [0, N)$, so $f$ is continuous. Hence each integral term on the left-hand side is differentiable for any $x \in [0, N)$, and note that $f$ is differentiable at every point. Dividing both sides of the eigen-equation by $\sqrt{x+\frac{1}{2}}$ and multiplying by $N^2+N$, we obtain

$$
-\int_0^x \sqrt{\xi+\frac{1}{2}} f(\xi) \mathrm{d}\xi + \int_x^N \sqrt{\xi+\frac{1}{2}} f(\xi) \mathrm{d}\xi = \lambda(N^2+N) \frac{f(x)}{\sqrt{x+\frac{1}{2}}}.
\tag{82}
$$

Differentiating the left-hand side with respect to $x$, we obtain

$$\frac{\mathrm{d}}{\mathrm{d}x}\left(-\int_0^x \sqrt{\xi+\tfrac{1}{2}}f(\xi)\mathrm{d}\xi + \int_x^N \sqrt{\xi+\tfrac{1}{2}}f(\xi)\mathrm{d}\xi\right) = -\sqrt{x+\frac{1}{2}}f(x) - \sqrt{x+\frac{1}{2}}f(x)$$

$$= -2\sqrt{x+\frac{1}{2}}f(x), \tag{83}$$

and differentiating the right-hand side gives

$$\lambda(N^2+N)\frac{\mathrm{d}}{\mathrm{d}x}\left(\frac{f(x)}{\sqrt{x+\frac{1}{2}}}\right) = \lambda(N^2+N)\left(\frac{f'(x)}{\sqrt{x+\frac{1}{2}}} - \frac{f(x)}{2\sqrt{x+\frac{1}{2}}\left(x+\frac{1}{2}\right)}\right). \tag{84}$$

Therefore, by comparing both sides of equation (82),

$$-2\sqrt{x+\frac{1}{2}}f(x) = \lambda(N^2+N)\left(\frac{f'(x)}{\sqrt{x+\frac{1}{2}}} - \frac{f(x)}{2\sqrt{x+\frac{1}{2}}\left(x+\frac{1}{2}\right)}\right) \tag{85}$$

is obtained. Rearranging yields the first-order ODE

$$f'(x) = \left(\frac{1}{2(x+\frac{1}{2})} - \frac{2(x+\frac{1}{2})}{\lambda(N^2+N)}\right)f(x), \tag{86}$$

and solving this yields

$$f(x) = C\sqrt{x+\frac{1}{2}}\exp\left(-\frac{x^2+x}{\lambda(N^2+N)}\right), \tag{87}$$

where $C \neq 0$ is a constant.

Substituting this into (82), we obtain

$$\int_0^x \sqrt{\xi+\frac{1}{2}}f(\xi)\,\mathrm{d}\xi = C\int_0^x (\xi+\tfrac{1}{2})\exp\left(-\frac{\xi^2+\xi}{\lambda(N^2+N)}\right)\mathrm{d}\xi$$

$$= \frac{C\lambda(N^2+N)}{2}\left(1-\exp\left(-\frac{x^2+x}{\lambda(N^2+N)}\right)\right) \tag{88}$$

and

$$\int_x^N \sqrt{\xi+\frac{1}{2}}f(\xi)\,\mathrm{d}\xi = \frac{C\lambda(N^2+N)}{2}\left(\exp\left(-\frac{x^2+x}{\lambda(N^2+N)}\right) - \exp\left(-\frac{N^2+N}{\lambda(N^2+N)}\right)\right)$$

$$= \frac{C\lambda(N^2+N)}{2}\left(\exp\left(-\frac{x^2+x}{\lambda(N^2+N)}\right) - \exp\left(-\frac{1}{\lambda}\right)\right). \tag{89}$$

Since the above holds, the left-hand side of the equation (87) can be written as follows.

$$-\int_0^x \sqrt{\xi+\frac{1}{2}}f(\xi)\mathrm{d}\xi + \int_x^N \sqrt{\xi+\frac{1}{2}}f(\xi)\mathrm{d}\xi$$

$$= -\frac{C\lambda(N^2+N)}{2}\left(1-\exp\left(-\frac{x^2+x}{\lambda(N^2+N)}\right)\right) + \frac{C\lambda(N^2+N)}{2}\left(\exp\left(-\frac{x^2+x}{\lambda(N^2+N)}\right) - \exp\left(-\frac{1}{\lambda}\right)\right) \tag{90}$$

$$= -\frac{C\lambda(N^2+N)}{2} - \frac{C\lambda(N^2+N)}{2}\exp\left(-\frac{1}{\lambda}\right) + C\lambda(N^2+N)\exp\left(-\frac{x^2+x}{\lambda(N^2+N)}\right)$$

On the other hand, the right-hand side of the equation (87) can be written as follows

$$\lambda(N^2+N)\frac{f(x)}{\sqrt{x+\frac{1}{2}}} = C\lambda(N^2+N)\exp\left(-\frac{x^2+x}{\lambda(N^2+N)}\right). \tag{91}$$

Therefore, we can eliminate $\exp\left(-\frac{x^2+x}{\lambda(N^2+N)}\right)$ and obtain

$$\exp\left(-\frac{1}{\lambda}\right) = -1. \tag{92}$$

Thus, using $k \in \mathbb{Z}$, we obtain

$$\frac{1}{\lambda} = (2k+1)i\pi \tag{93}$$

that is,

$$\lambda = \frac{1}{(2k+1)i\pi}. \tag{94}$$

This establishes necessity for being an eigenvalue. Sufficiency can be shown by substituting the eigenvalue-eigenvector pair obtained above into the eigen-equation. $\qquad\square$

Similarly, we can obtain the eigenvalues of $\mathcal{A}^{\text{skew}}$.

*Proof of Proposition 5.2.* From the definition, it can be written as

$$\mathcal{A}^{\text{skew}} = (N^2 + N)\mathcal{S}^{(N)}, \tag{95}$$

and from Proposition E.3, the eigenvalues of $\mathcal{S}^{(N)}$ are written as

$$\frac{1}{(2k+1)i\pi}. \tag{96}$$

Thus, the eigenvalues of $\mathcal{A}^{\text{skew}}$ can be written as

$$\frac{N^2 + N}{(2k+1)i\pi}. \tag{97}$$

$\qquad\square$

Using the theorem above, we can identify the limit of the eigenvalues of the scaled discrete operator $\mathcal{S}_\Delta$ as follows.

**Theorem E.4.** *Noting that the eigenvalues of $\mathcal{S}_\Delta^{(N)}$ are purely imaginary. Among them, let $iv_n^{(N)}$ denote the $n$-th largest eigenvalue whose imaginary part is positive. Then, for any $n \in \mathbb{Z}_{\geq 1}$,*

$$\lim_{N\to\infty} v_n^{(N)} = \frac{1}{(2n-1)\pi} \tag{98}$$

*holds.*

*Proof.* By Proposition E.2, $\left\|\mathcal{S}^{(N)} - \mathcal{S}_\Delta^{(N)}\right\|_{\mathcal{L}(L^2(\mathbb{R}), L^2(\mathbb{R}))}$ converges to 0. Moreover, since $\mathcal{S}^{(N)}, \mathcal{S}_\Delta^{(N)}$ are skew-symmetric and compact, $i\mathcal{S}^{(N)}, i\mathcal{S}_\Delta^{(N)}$ are self-adjoint and compact. Therefore, by Proposition E.3, noting that the $n$-th largest positive eigenvalue of $i\mathcal{S}^{(N)}$ is $(2n-1)^{-1}\pi^{-1}$, Weyl's inequality implies

$$\begin{aligned}
0 &\leq \left|v_n^{(N)} - \frac{1}{(2n-1)\pi}\right| \\
&= \left|iv_n^{(N)} - i\frac{1}{(2n-1)\pi}\right| \\
&\leq \left\|i\mathcal{S}^{(N)} - i\mathcal{S}_\Delta^{(N)}\right\|_{\mathcal{L}(L^2(\mathbb{R},\mathbb{C}), L^2(\mathbb{R},\mathbb{C}))} \\
&= \left\|\mathcal{S}^{(N)} - \mathcal{S}_\Delta^{(N)}\right\|_{\mathcal{L}(L^2(\mathbb{R}), L^2(\mathbb{R}))} \xrightarrow{N\to\infty} 0.
\end{aligned} \tag{99}$$

Hence,

$$\lim_{N\to\infty}\left|v_n^{(N)} - \frac{1}{(2n-1)\pi}\right| = 0,\tag{100}$$

and, then

$$\lim_{N\to\infty} v_n^{(N)} = \frac{1}{(2n-1)\pi}\tag{101}$$

hold. □

Using this, we obtain the proof of Theorem 5.4.

*Proof of Theorem 5.4.* By definition, among the eigenvalues of $A^{\mathrm{skew}} \in \mathbb{R}^{N\times N}$, the $n$-th largest one whose imaginary part is positive is $\tau_n^{(N)}$. By Lemma E.1, this implies that the $n$-th largest positive imaginary part among the eigenvalues of the operator $\mathcal{A}^{\mathrm{skew}}$ is also $\tau_n^{(N)}$. From the definition, it holds that

$$\mathcal{A}^{\mathrm{skew}} = (N^2 + N)\mathcal{S}^{(N)}\tag{102}$$

and therefore $\tau_n^{(N)}$ and $v_n^{(N)}$ in Proposition E.3 satisfy the following relation.

$$\tau_n^{(N)} = (N^2 + N)v_n^{(N)}\tag{103}$$

Hence,

$$\lim_{N\to\infty}\frac{\tau_n^{(N)}}{N^2} = \lim_{N\to\infty}\frac{N^2+N}{N^2}v_n^{(N)} = 1\cdot\lim_{N\to\infty} v_n^{(N)} = \frac{1}{(2n-1)\pi}\tag{104}$$

is obtained. □

## F. Proof of Theorem 5.5

For notational simplicity, in what follows the $N \times N$ matrix $A^{\mathrm{skew}}$ is written using $\chi_0(x) = \sqrt{x + \frac{1}{2}}$ as

$$A_{k,l}^{\mathrm{skew}} = -\chi_0(k)\chi_0(l)\operatorname{sign}(k - l).\tag{105}$$

With this notation, the following holds for the eigenvalues of $A^{\mathrm{skew}}$.

**Lemma F.1.** *Noting that the eigenvalues of $A^{\mathrm{skew}}$ are purely imaginary. For an eigenvalue $\lambda = i\tau$,*

$$\prod_{k=0}^{N-1}\frac{|\chi_0(k)|^2 - \lambda}{|\chi_0(k)|^2 + \lambda} = (-1)^{N+1}\tag{106}$$

*holds. Moreover, let $\tau_n$ denote the $n$-th largest imaginary part among positive imaginary part of the eigenvalues. Then,*

$$\sum_{k=0}^{N-1}\arctan\left(\frac{|\chi_0(k)|^2}{\tau_n}\right) = -\frac{\pi}{2} + \pi n\tag{107}$$

*is satisfied.*

*Proof.* Consider the following eigenvalue equation.

$$A^{\mathrm{skew}} f = \lambda f\tag{108}$$

Writing this componentwise gives

$$-\chi_0(k)\sum_{l=0}^{k-1}\chi_0(l)f_l + \chi_0(k)\sum_{l=k+1}^{N-1}\chi_0(l)f_l = \lambda f_k.\tag{109}$$

Dividing both sides by $\chi_0(k)$ yields

$$-\sum_{l=0}^{k-1}\chi_0(l)f_l + \sum_{l=k+1}^{N-1}\chi_0(l)f_l = \lambda\frac{f_k}{\chi_0(k)} \tag{110}$$

and similarly, replacing $k$ by $k+1$ gives

$$-\sum_{l=0}^{k}\chi_0(l)f_l + \sum_{l=k+2}^{N-1}\chi_0(l)f_l = \lambda\frac{f_{k+1}}{\chi_0(k+1)} \tag{111}$$

Taking the difference of these two equations, we obtain

$$-\chi_0(k+1)f_{k+1} - \chi_0(k)f_k = \lambda\left(\frac{f_{k+1}}{\chi_0(k+1)} - \frac{f_k}{\chi_0(k)}\right) \tag{112}$$

Rearranging this gives

$$-\left(\chi_0(k+1) + \frac{\lambda}{\chi_0(k+1)}\right)f_{k+1} = \left(\chi_0(k) - \frac{\lambda}{\chi_0(k)}\right)f_k \tag{113}$$

and hence we obtain the following recurrence relation for the eigenvector.

$$
\begin{aligned}
f_{k+1} &= -\frac{\chi_0(k) - \frac{\lambda}{\chi_0(k)}}{\chi_0(k+1) + \frac{\lambda}{\chi_0(k+1)}}f_k \\
&= -\frac{\chi_0(k+1)}{\chi_0(k)}\frac{|\chi_0(k)|^2 - \lambda}{|\chi_0(k+1)|^2 + \lambda}f_i
\end{aligned}
\tag{114}
$$

Therefore, for each $1 \leq j \leq N-1$, expressing $f_j$ in terms of the first term $f_0$ yields the following.

$$
\begin{aligned}
f_i &= f_0 \prod_{k=0}^{i-1}\left(-\frac{\chi_0(k+1)}{\chi_0(k)}\frac{\chi_0(k)\chi_0(k) - \lambda}{\chi_0(k+1)\chi_0(k+1) + \lambda}\right) \\
&= f_0(-1)^i\left(\prod_{k=0}^{i-1}\frac{\chi_0(k+1)}{\chi_0(k)}\right)\left(\prod_{k=0}^{i-1}\frac{\chi_0(k)\chi_0(k) - \lambda}{\chi_0(k+1)\chi_0(k+1) + \lambda}\right) \\
&= f_0(-1)^i\frac{\chi_0(i)}{\chi_0(0)}\prod_{k=0}^{i-1}\frac{|\chi_0(k)|^2 - \lambda}{|\chi_0(k+1)|^2 + \lambda} \\
&= f_0\frac{\chi_0(i)}{\chi_0(0)}P_{i-1}
\end{aligned}
\tag{115}
$$

where

$$P_i := (-1)^{i+1}\prod_{k=0}^{i}\frac{|\chi_0(k)|^2 - \lambda}{|\chi_0(k+1)|^2 + \lambda}. \tag{116}$$

Note that if $f_0 = 0$, then $f$ becomes the zero vector and is no longer an eigenvector. Now, set $C_l = \left(|\chi_0(l+1)|^2 + \lambda\right)P_l$. Then, for $1 \leq j \leq N-1$, the following hold:

$$C_l - C_{l-1} = \left(|\chi_0(l+1)|^2 + \lambda\right) P_l - \left(|\chi_0(l)|^2 + \lambda\right) P_{l-1}$$

$$= \left(|\chi_0(l+1)|^2 + \lambda\right)(-1)^{l+1} \prod_{m=0}^{l} \frac{|\chi_0(m)|^2 - \lambda}{|\chi_0(m+1)|^2 + \lambda}$$

$$- \left(|\chi_0(l)|^2 + \lambda\right)(-1)^{l} \prod_{m=0}^{l-1} \frac{|\chi_0(m)|^2 - \lambda}{|\chi_0(m+1)|^2 + \lambda}$$

$$= -\left(\left(|\chi_0(l)|^2 - \lambda\right) + \left(|\chi_0(l)|^2 + \lambda\right)\right)(-1)^{l} \prod_{m=0}^{l-1} \frac{|\chi_0(m)|^2 - \lambda}{|\chi_0(m+1)|^2 + \lambda} \tag{117}$$

$$= -2|\chi_0(l)|^2 (-1)^{l} \prod_{m=0}^{l-1} \frac{|\chi_0(m)|^2 - \lambda}{|\chi_0(m+1)|^2 + \lambda}$$

$$= -2|\chi_0(l)|^2 P_{l-1}$$

and

$$C_l + C_{l-1} = \left(|\chi_0(l+1)|^2 + \lambda\right) P_l + \left(|\chi_0(l)|^2 + \lambda\right) P_{l-1}$$

$$= \left(|\chi_0(l+1)|^2 + \lambda\right)(-1)^{l+1} \prod_{m=0}^{l} \frac{|\chi_0(m)|^2 - \lambda}{|\chi_0(m+1)|^2 + \lambda}$$

$$+ \left(|\chi_0(l)|^2 + \lambda\right)(-1)^{l} \prod_{m=0}^{l-1} \frac{|\chi_0(m)|^2 - \lambda}{|\chi_0(m+1)|^2 + \lambda} \tag{118}$$

$$= -\left(\left(|\chi_0(l)|^2 - \lambda\right) - \left(|\chi_0(l)|^2 + \lambda\right)\right)(-1)^{l} \prod_{m=0}^{l-1} \frac{|\chi_0(m)|^2 - \lambda}{|\chi_0(m+1)|^2 + \lambda}$$

$$= 2\lambda(-1)^{l} \prod_{m=0}^{l-1} \frac{|\chi_0(m)|^2 - \lambda}{|\chi_0(m+1)|^2 + \lambda}$$

$$= 2\lambda P_{l-1}.$$

In particular,

$$|\chi_0(l)|^2 P_{l-1} = -\frac{C_l - C_{l-1}}{2} \tag{119}$$

and

$$\lambda P_{l-1} = \frac{C_l + C_{l-1}}{2} \tag{120}$$

can be written. Substituting this into the eigenvalue equation (108) yields the following.

$$0 = -\sum_{l=0}^{k-1} \chi_0(l) f_l + \sum_{l=k+1}^{N-1} \chi_0(l) f_l - \lambda \frac{f_k}{\chi_0(k)}$$

$$= -f_0 \chi_0(0) - \sum_{l=1}^{k-1} \chi_0(l) \frac{f_0 \chi_0(l)}{\chi_0(0)} P_{l-1} + f_0 \sum_{l=k+1}^{N-1} \chi_0(l) \frac{f_0 \chi_0(l)}{\chi_0(0)} P_{l-1} - \lambda \frac{1}{\chi_0(k)} \frac{f_0 \chi_0(k)}{\chi_0(0)} P_{k-1}$$

$$= \frac{f_0}{\chi_0(0)} \left(-|\chi_0(0)|^2 - \sum_{l=1}^{k-1} |\chi_0(l)|^2 P_{l-1} + \sum_{l=k+1}^{N-1} |\chi_0(l)|^2 P_{l-1} - \lambda P_{k-1}\right) \tag{121}$$

$$= \frac{f_0}{\chi_0(0)} \left(-|\chi_0(0)|^2 + \sum_{l=1}^{k-1} \frac{C_l - C_{l-1}}{2} - \sum_{l=k+1}^{N-1} \frac{C_l - C_{l-1}}{2} - \frac{C_k + C_{k-1}}{2}\right)$$

$$= \frac{f_0}{\chi_0(0)} \left(-|\chi_0(0)|^2 + \frac{C_{k-1} - C_0}{2} - \frac{C_{N-1} - C_k}{2} - \frac{C_k + C_{k-1}}{2}\right)$$

$$= \frac{f_0}{\chi_0(0)} \left(-|\chi_0(0)|^2 - \frac{C_{N-1} + C_0}{2}\right)$$

In particular, substituting

$$C_0 = \left(|\chi_0(1)|^2 + \lambda\right) P_0 \tag{122}$$

$$= \left(|\chi_0(1)|^2 + \lambda\right) \left(-\frac{|\chi_0(0)|^2 - \lambda}{|\chi_0(1)|^2 + \lambda}\right) \tag{123}$$

$$= -\left(|\chi_0(0)|^2 - \lambda\right) \tag{124}$$

and

$$C_{N-1} = \left(|\chi_0(N)|^2 + \lambda\right) P_{N-1} \tag{125}$$

$$= \left(|\chi_0(N)|^2 + \lambda\right)(-1)^N \prod_{k=0}^{N-1} \frac{|\chi_0(k)|^2 - \lambda}{|\chi_0(k+1)|^2 + \lambda}, \tag{126}$$

it holds that

$$
\begin{aligned}
0 &= \frac{f_0}{\chi_0(0)} \left(-|\chi_0(0)|^2 - \frac{C_{N-1} - C_0}{2}\right) \\
&= \frac{f_0}{\chi_0(0)} \left(-|\chi_0(0)|^2 - \frac{1}{2}\left(\left(|\chi_0(N)|^2 + \lambda\right)(-1)^N \prod_{k=0}^{N-1} \frac{|\chi_0(k)|^2 - \lambda}{|\chi_0(k+1)|^2 + \lambda} - \left(|\chi_0(0)|^2 - \lambda\right)\right)\right) \\
&= \frac{f_0}{2\chi_0(0)} \left(-|\chi_0(0)|^2 + \lambda - \left(|\chi_0(N)|^2 + \lambda\right)(-1)^N \prod_{k=0}^{N-1} \frac{|\chi_0(k)|^2 - \lambda}{|\chi_0(k+1)|^2 + \lambda}\right).
\end{aligned}
\tag{127}
$$

Using $f_0 \neq 0$ and simplifying, we obtain

$$\prod_{k=0}^{N-1} \frac{|\chi_0(k)|^2 - \lambda}{|\chi_0(k+1)|^2 + \lambda} = -(-1)^N \frac{|\chi_0(0)|^2 - \lambda}{|\chi_0(N)|^2 + \lambda}, \tag{128}$$

that is,

$$\prod_{k=0}^{N-1} \frac{|\chi_0(k)|^2 - \lambda}{|\chi_0(k)|^2 + \lambda} = \frac{|\chi_0(N)|^2 + \lambda}{|\chi_0(0)|^2 - \lambda} \prod_{k=0}^{N-1} \frac{|\chi_0(k)|^2 - \lambda}{|\chi_0(k+1)|^2 + \lambda} = (-1)^{N+1} \tag{129}$$

holds. Next we prove (107). For an arbitrary eigenvalue $i\tau$,

$$\prod_{k=0}^{N-1} \frac{|\chi_0(k)|^2 - i\tau}{|\chi_0(k)|^2 + i\tau} = (-1)^{N+1} \tag{130}$$

holds. We compare the polar forms of both sides. The absolute values are

$$\left|\frac{|\chi_0(k)|^2 - i\tau}{|\chi_0(k)|^2 + i\tau}\right| = 1 \tag{131}$$

so they are always equal. We now consider the arguments. For each $k$, using integers $m_k$, it can be written as

$$
\begin{aligned}
\arg\left(\frac{|\chi_0(k)|^2 - i\tau}{|\chi_0(k)|^2 + i\tau}\right) &= \arg\left(|\chi_0(k)|^2 - i\tau\right) - \arg\left(|\chi_0(k)|^2 + i\tau\right) + 2\pi m_1 \\
&= 2\arg\left(|\chi_0(k)|^2 - i\tau\right) + 2\pi m_2 \\
&= -2\arctan\left(\frac{\tau}{|\chi_0(k)|^2}\right) + 2\pi m_2,
\end{aligned}
\tag{132}
$$

and thus the argument of the left-hand side can be written as

$$
\arg\left(\prod_{k=0}^{N-1}\frac{|\chi_0(k)|^2 - i\tau}{|\chi_0(k)|^2 + i\tau}\right) = \sum_{k=0}^{N-1}\arg\left(\frac{|\chi_0(k)|^2 - i\tau}{|\chi_0(k)|^2 + i\tau}\right) + 2\pi m_3
$$

$$
= -2\sum_{k=0}^{N-1}\arctan\left(\frac{\tau}{|\chi_0(k)|^2}\right) + 2\pi m_4.
$$

(133)

On the other hand, the argument of the right-hand side can be written as

$$
\arg\left((-1)^{N+1}\right) = (N+1)\arg(-1) + 2\pi m_5
$$

$$
= (N+1)\pi + 2\pi m_5.
$$

(134)

Since (133) and (134) are equal, we obtain the following form:

$$
-2\sum_{k=0}^{N-1}\arctan\left(\frac{\tau}{|\chi_0(k)|^2}\right) + 2\pi m_4 = (N+1)\pi + 2\pi m_5,
$$

(135)

that is,

$$
\sum_{k=0}^{N-1}\arctan\left(\frac{\tau}{|\chi_0(k)|^2}\right) = -\frac{N+1}{2}\pi + \pi m_6.
$$

(136)

Using the functional identity

$$
\arctan\left(\frac{1}{x}\right) = \frac{\pi}{2} - \arctan(x)
$$

(137)

we can rewrite the left-hand side of the equation (135) as

$$
\sum_{k=0}^{N-1}\arctan\left(\frac{\tau}{|\chi_0(k)|^2}\right) = \sum_{k=0}^{N-1}\left(\frac{\pi}{2} - \arctan\left(\frac{|\chi_0(k)|^2}{\tau}\right)\right)
$$

$$
= \frac{N\pi}{2} - \sum_{k=0}^{N-1}\arctan\left(\frac{|\chi_0(k)|^2}{\tau}\right).
$$

(138)

Substituting this into (136) yields

$$
\frac{N\pi}{2} - \sum_{k=0}^{N-1}\arctan\left(\frac{|\chi_0(k)|^2}{\tau}\right) = -\frac{N+1}{2}\pi + \pi m_6
$$

(139)

that is,

$$
\sum_{k=0}^{N-1}\arctan\left(\frac{|\chi_0(k)|^2}{\tau}\right) = (N+1)\pi - \frac{1}{2}\pi + \pi m_7.
$$

(140)

In particular, if we set $n = m_7 + N + 1$, it can be written as

$$
\sum_{k=0}^{N-1}\arctan\left(\frac{|\chi_0(k)|^2}{\tau}\right) = -\frac{1}{2}\pi + \pi n
$$

(141)

Up to this point, for any positive imaginary part $\tau > 0$, we have shown that it can be expressed by (141). Next, consider ordering the positive imaginary parts $\tau > 0$ by magnitude, that is, consider the sequence $\{\tau_k\}_{k=1,\ldots,\lfloor\frac{N}{2}\rfloor}$ arranging imaginary parts $\tau > 0$ in monotonically decreasing order.

Now, considering the range of the left-hand side of (141), since inequality $0 < \arctan(x) < \frac{\pi}{2}$ holds for $x > 0$, we obtain

$$
0 < \sum_{k=0}^{N-1}\arctan\left(\frac{|\chi_0(k)|^2}{\tau}\right) < \frac{N\pi}{2}.
$$

(142)

Substituting the right-hand side of (141) gives

$$0 < -\frac{1}{2}\pi + \pi n < \frac{N\pi}{2}. \tag{143}$$

Therefore, as a necessary condition for (141) to hold, we obtain $n = 1, \ldots, \lfloor \frac{N}{2} \rfloor$. Noting that $\arctan(x)$ is monotonically increasing, the left-hand side of (141) is monotonically decreasing in $\tau$. Thus, for the integer $n$ chosen on the right-hand side of (141), the equality (141) corresponds to the $n$-th largest positive imaginary part $\tau_n > 0$.

$\square$

Using this lemma together with Theorem 5.4, we can prove Theorem 5.5.

*proof of theorem 5.5.* Noting Theorem 5.4, our goal is to prove that for the $n$-th largest positive imaginary part $\tau_n^{(N)}$ of the $N \times N$ matrix $A^{\text{skew}}$, the deviation after subtracting the leading term

$$c_n^{(N)} = \tau_n^{(N)} - \frac{N^2}{(2n-1)\pi} \tag{144}$$

converges, i.e., that the limit

$$\lim_{N \to \infty} c_n^{(N)} = \lim_{N \to \infty} \left( \tau_n^{(N)} - \frac{N^2}{(2n-1)\pi} \right) \tag{145}$$

converges to a constant. Keeping this in mind, we proceed with the proof.

From Lemma F.1,

$$\sum_{k=0}^{N-1} \arctan\left( \frac{k + \frac{1}{2}}{\tau_n^{(N)}} \right) = -\frac{\pi}{2} + \pi n \tag{146}$$

holds. Here note that $|\chi_0(k)|^2 = k + \frac{1}{2}$. We now take a Taylor expansion of the left-hand side. As $N \to \infty$, $\tau_n^{(N)} \sim \frac{N^2}{(2n-1)\pi}$, so for sufficiently large $N$, for $k = 0, \ldots, N-1$,

$$\frac{k + \frac{1}{2}}{\tau_n^N} \ll 1 \tag{147}$$

holds. Therefore, for $|x| < 1$, the Taylor expansion of $\arctan(x)$,

$$\arctan(x) = \sum_{l=0}^{\infty} \frac{(-1)^l}{2l+1} x^{2l+1} \tag{148}$$

implies the following.

$$
\begin{aligned}
&\sum_{k=0}^{N-1} \arctan\left( \frac{k + \frac{1}{2}}{\tau_n^{(N)}} \right) \\
&= \sum_{k=0}^{N-1} \left( \sum_{l=0}^{\infty} \frac{(-1)^l}{2l+1} \left( \frac{k + \frac{1}{2}}{\tau_n^{(N)}} \right)^{2l+1} \right) \\
&= \sum_{l=0}^{\infty} \frac{(-1)^l}{2l+1} \frac{1}{\left( \tau_n^{(N)} \right)^{2l+1}} \sum_{k=0}^{N-1} \left( k + \frac{1}{2} \right)^{2l+1}
\end{aligned}
$$

$$= \frac{1}{\tau_n^{(N)}} \sum_{k=0}^{N-1} \left(k + \frac{1}{2}\right) - \frac{1}{3} \frac{1}{\left(\tau_n^{(N)}\right)^3} \sum_{k=0}^{N-1} \left(k + \frac{1}{2}\right)^3 + \sum_{l=2}^{\infty} \frac{(-1)^l}{2l+1} \frac{1}{\left(\tau_n^{(N)}\right)^{2l+1}} \sum_{k=0}^{N-1} \left(k + \frac{1}{2}\right)^{2l+1}$$

$$= \frac{1}{\tau_n^{(N)}} \frac{N^2}{2} - \frac{1}{3} \frac{1}{\left(\tau_n^{(N)}\right)^3} \left(-\frac{N^2}{8} + \frac{N^4}{4}\right) + \sum_{l=2}^{\infty} \frac{(-1)^l}{2l+1} \frac{1}{\left(\tau_n^{(N)}\right)^{2l+1}} \sum_{k=0}^{N-1} \left(k + \frac{1}{2}\right)^{2l+1} \tag{149}$$

$$= \frac{N^2}{2\tau_n^{(N)}} - \frac{1}{3} \frac{1}{\left(\tau_n^{(N)}\right)^3} \left(-\frac{N^2}{8} + \frac{N^4}{4}\right) + o\left(\frac{1}{N^2}\right)$$

Here, the order estimate of the final third term follows as below. If $l \geq 2$, since $\tau_n^{(N)} = \Theta(N^2)$, the numerator satisfies $\left(\tau_n^{(N)}\right)^{2l+1} = \Theta(N^{4l+2})$, and the denominator satisfies

$$\sum_{k=0}^{N-1} \left(k + \frac{1}{2}\right)^{2l+1} = \Theta\left(N^{2l+2}\right) \tag{150}$$

hence

$$\frac{1}{\left(\tau_n^{(N)}\right)^{2l+1}} \sum_{k=0}^{N-1} \left(k + \frac{1}{2}\right)^{2l+1} = \Theta\left(N^{2l-4l}\right) = \Theta\left(N^{-2l}\right) < o(N^{-2}) \tag{151}$$

and the sum is also $o(N^{-2})$.

Returning to the expression (149) and using the fact that

$$\frac{1}{x} = \frac{1}{1 - (1 - x)}$$

$$= \sum_{k=0}^{\infty} (1 - x)^k$$

$$= 1 + (1 - x) + \sum_{k=2}^{\infty} (1 - x)^k \tag{152}$$

$$= 1 + (1 - x) + (1 - x)^2 \sum_{k=0}^{\infty} (1 - x)^k$$

$$= 1 + (1 - x) + \frac{(1 - x)^2}{x}$$

identity holds, we expand the first term of the equation (149). This yields

$$\frac{1}{\tau_n^{(N)}} = \frac{(2n - 1)\pi}{N^2} \frac{1}{\frac{(2n-1)\pi}{N^2} \tau_n^{(N)}}$$

$$= \frac{(2n - 1)\pi}{N^2} \left(1 + \left(1 - \frac{(2n - 1)\pi}{N^2} \tau_n^{(N)}\right) + \frac{N^2}{(2n - 1)\pi \tau_n^{(N)}} \left(1 - \frac{(2n - 1)\pi}{N^2} \tau_n^{(N)}\right)^2.\right) \tag{153}$$

Therefore, the first term of (149) can be written as

$$\frac{N^2}{2\tau_n^N} = \left(n\pi - \frac{\pi}{2}\right) \left(1 + \left(1 - \frac{(2n - 1)\pi}{N^2} \tau_n^{(N)}\right) + \frac{N^2}{(2n - 1)\pi \tau_n^{(N)}} \left(1 - \frac{(2n - 1)\pi}{N^2} \tau_n^{(N)}\right)^2\right) \tag{154}$$

Substituting (149) and (154) into the condition (146), we obtain

$$
\begin{aligned}
-\frac{\pi}{2} + \pi n = {}& \left( n\pi - \frac{\pi}{2} \right) \left( 1 + \left( 1 - \frac{(2n-1)\pi}{N^2} \tau_n^{(N)} \right) + \frac{N^2}{(2n-1)\pi\tau_n^{(N)}} \left( 1 - \frac{(2n-1)\pi}{N^2} \tau_n^{(N)} \right)^2 \right) \\
& - \frac{1}{3} \frac{1}{\left( \tau_n^{(N)} \right)^3} \left( -\frac{N^2}{8} + \frac{N^4}{4} \right) + o\left( \frac{1}{N^2} \right)
\end{aligned}
\tag{155}
$$

Rearranging terms gives

$$
\begin{aligned}
& \left( -\frac{\pi}{2} + \pi n \right) - \left( n\pi - \frac{\pi}{2} \right) \left( 1 + \left( 1 - \frac{(2n-1)\pi}{N^2} \tau_n^{(N)} \right) \right) \\
& = \frac{N^2}{(2n-1)\pi\tau_n^{(N)}} \left( n\pi - \frac{\pi}{2} \right) \left( 1 - \frac{(2n-1)\pi}{N^2} \tau_n^{(N)} \right)^2 - \frac{1}{3} \frac{1}{\left( \tau_n^{(N)} \right)^3} \left( -\frac{N^2}{8} + \frac{N^4}{4} \right) + o\left( \frac{1}{N^2} \right).
\end{aligned}
\tag{156}
$$

Multiplying both sides by $\frac{2N^2}{(2n-1)^2\pi^2}$ and simplifying yields the following.

$$
\begin{aligned}
\tau_n^{(N)} - \frac{N^2}{(2n-1)\pi} = {}& \left( \frac{N^2}{(2n-1)\pi} \right)^2 \frac{1}{\tau_n^{(N)}} \left( 1 - \frac{(2n-1)\pi}{N^2} \tau_n^{(N)} \right)^2 \\
& - \frac{1}{3} \frac{2N^2}{(2n-1)^2\pi^2} \frac{1}{\left( \tau_n^{(N)} \right)^3} \left( -\frac{N^2}{8} + \frac{N^4}{4} \right) + o\left( 1 \right) \\
= {}& \frac{1}{\tau_n^{(N)}} \left( \tau_n^{(N)} - \frac{N^2}{(2n-1)\pi} \right)^2 \\
& - \frac{1}{3} \frac{2N^2}{(2n-1)^2\pi^2} \frac{1}{\left( \tau_n^{(N)} \right)^3} \left( -\frac{N^2}{8} + \frac{N^4}{4} \right) + o\left( 1 \right)
\end{aligned}
\tag{157}
$$

Writing this in terms of $c_n^{(N)}$, we obtain

$$
c_n^{(N)} = \frac{\left( c_n^{(N)} \right)^2}{\tau_n^{(N)}} - \frac{1}{3} \frac{2N^2}{(2n-1)^2\pi^2} \frac{1}{\left( \tau_n^{(N)} \right)^3} \left( -\frac{N^2}{8} + \frac{N^4}{4} \right) + o\left( 1 \right).
\tag{158}
$$

Moving terms and simplifying,

$$
c_n^{(N)} \left( 1 - \frac{c_n^{(N)}}{\tau_n^{(N)}} \right) = -\frac{2(2n-1)\pi}{3} \left( \frac{N^2}{(2n-1)\pi} \frac{1}{\tau_n^{(N)}} \right)^3 \left( \frac{1}{4} - \frac{1}{8N^2} \right) + o\left( 1 \right)
\tag{159}
$$

holds. Noting that

$$
\lim_{N\to\infty} \frac{c_n^{(N)}}{\tau_n^{(N)}} = \lim_{N\to\infty} \left( 1 - \frac{N^2}{(2n-1)\pi\tau_n} \right) = 1 - 1 = 0
\tag{160}
$$

holds, we obtain the limit of $c_n^{(N)}$:

$$
\begin{aligned}
\lim_{N\to\infty} c_n^{(N)} &= \lim_{N\to\infty} c_n^{(N)} \left( 1 - \frac{c_n^{(N)}}{\tau_n^{(N)}} \right) \\
&= -\frac{2(2n-1)\pi}{3} \lim_{N\to\infty} \left( \frac{N^2}{(2n-1)\pi} \frac{1}{\tau_n^{(N)}} \left( \frac{1}{4} - \frac{1}{8N^2} \right) \right)^3 \\
&= -\frac{2(2n-1)\pi}{3} \cdot 1^3 \left( \frac{1}{4} - 0 \right) \\
&= -\frac{(2n-1)\pi}{6}
\end{aligned}
\tag{161}
$$

as conclusion. $\qquad\square$

## G. The Proof of Theorem 6.1, Theorem 6.4, Theorem 6.6

In this section we derive an explicit solution of the LSSL based on parametric CHIP.

*Proof of Lemma 6.5.* Since $\varphi, \psi, g \in L^2([0, N], \mathbb{C})$, $\varphi\psi, \varphi g \in L^1([0, N], \mathbb{C})$ holds. Therefore,

$$p(s_1, ..., s_n) := \left( \prod_{i=0}^{n-1} \varphi(s_i)\psi(s_i) \right) \psi(s_n)g(s_n) \tag{162}$$

is integrable on $[0, N]^n$ by the Fubini–Tonelli theorem.

Hence, $p$ is integrable on $\{(s_1, ..., s_n) \mid a \leq s_n \leq s_{n-1} \leq \cdots \leq s_1 \leq x\}$, and by Fubini's theorem the following holds.

$$
\begin{aligned}
&\mathcal{K}_{\varphi,\psi}^n[g](x) \\
&= \varphi(x) \int_0^x \psi(s_1)\varphi(s_1) \int_0^{s_1} \psi(s_2) \ldots \varphi(s_{n-1}) \int_0^{s_{n-1}} \psi(s_n)g(s_n) \mathrm{d}s_n \mathrm{d}s_{n-1} \ldots \mathrm{d}s_2 \mathrm{d}s_1 \\
&= \varphi(x) \int_0^x \int_0^{s_1} \cdots \int_0^{s_{n-1}} \left( \prod_{i=0}^{n-1} \varphi(s_i)\psi(s_i) \right) \psi(s_n)g(s_n) \mathrm{d}s_n \mathrm{d}s_{n-1} \ldots \mathrm{d}s_2 \mathrm{d}s_1 \\
&= \varphi(x) \int_{0 \leq s_n \leq s_{n-1} \leq \cdots \leq s_1 \leq x} \left( \prod_{i=0}^{n-1} \varphi(s_i)\psi(s_i) \right) \psi(s_n)g(s_n) \mathrm{d}s_n \mathrm{d}s_{n-1} \cdots \mathrm{d}s_2 \mathrm{d}s_1 \\
&= \varphi(x) \int_0^x \psi(s_n)g(s_n) \left( \int_{s_n \leq s_{n-1} \leq \cdots \leq s_1 \leq x} \left( \prod_{i=0}^{n-1} \varphi(s_i)\psi(s_i) \right) \mathrm{d}s_{n-1} \cdots \mathrm{d}s_2 \mathrm{d}s_1 \right) \mathrm{d}s_n \\
&= \varphi(x) \int_0^x \psi(\xi)g(\xi) \left( \int_{\xi \leq s_{n-1} \leq \cdots \leq s_1 \leq x} \left( \prod_{i=0}^{n-1} \varphi(s_i)\psi(s_i) \right) \mathrm{d}s_{n-1} \cdots \mathrm{d}s_2 \mathrm{d}s_1 \right) \mathrm{d}\xi
\end{aligned}
\tag{163}
$$

Here, Compute the multiple integral inside the outer integral

$$I_{\xi,x} := \int_{\xi \leq s_{n-1} \leq \cdots \leq s_1 \leq x} \left( \prod_{i=0}^{n-1} \varphi(s_i)\psi(s_i) \right) \mathrm{d}s_{n-1} \cdots \mathrm{d}s_2 \mathrm{d}s_1. \tag{164}$$

For any $\sigma \in S_{n-1}$, using

$$\prod_{i=0}^{n-1} \varphi(s_i)\psi(s_i) = \prod_{i=0}^{n-1} \varphi\left(s_{\sigma(i)}\right) \psi\left(s_{\sigma(i)}\right), \tag{165}$$

it hold that

$$
\begin{aligned}
&\int_{[\xi,x]^{n-1}} \left( \prod_{i=0}^{n-1} \varphi(s_i)\psi(s_i) \right) \mathrm{d}s_{n-1} \cdots \mathrm{d}s_2 \mathrm{d}s_1 \\
&= \sum_{\sigma \in S_{n-1}} \int_{\xi \leq s_{\sigma(n-1)} \leq \cdots \leq s_{\sigma(1)} \leq x} \left( \prod_{i=0}^{n-1} \varphi(s_i)\psi(s_i) \right) \mathrm{d}s_{n-1} \cdots \mathrm{d}s_2 \mathrm{d}s_1 \\
&= \sum_{\sigma \in S_{n-1}} \int_{\xi \leq s_{\sigma(n-1)} \leq \cdots \leq s_{\sigma(1)} \leq x} \left( \prod_{i=0}^{n-1} \varphi\left(s_{\sigma(i)}\right) \psi\left(s_{\sigma(i)}\right) \right) \mathrm{d}s_{n-1} \cdots \mathrm{d}s_2 \mathrm{d}s_1
\end{aligned}
$$

$$
\begin{aligned}
&= \sum_{\sigma \in S_{n-1}} \int_{\xi \le s_{n-1} \le \cdots \le s_1 \le x} \left( \prod_{i=0}^{n-1} \varphi(s_i)\psi(s_i) \right) \mathrm{d}s_{n-1} \cdots \mathrm{d}s_2 \mathrm{d}s_1 \\
&= |S_{n-1}| \int_{\xi \le s_{n-1} \le \cdots \le s_1 \le x} \left( \prod_{i=0}^{n-1} \varphi(s_i)\psi(s_i) \right) \mathrm{d}s_{n-1} \cdots \mathrm{d}s_2 \mathrm{d}s_1 \\
&= (n-1)! \int_{\xi \le s_{n-1} \le \cdots \le s_1 \le x} \left( \prod_{i=0}^{n-1} \varphi(s_i)\psi(s_i) \right) \mathrm{d}s_{n-1} \cdots \mathrm{d}s_2 \mathrm{d}s_1 \\
&= (n-1)! I_{\xi,x}.
\end{aligned}
\tag{166}
$$

On the other hand, by separating variables, we can compute

$$
\int_{[\xi,x]^{n-1}} \left( \prod_{i=0}^{n-1} \varphi(s_i)\psi(s_i) \right) \mathrm{d}s_{n-1} \cdots \mathrm{d}s_2 \mathrm{d}s_1 = \int_{\xi}^{x} \varphi(s_{n-1})\psi(s_{n-1}) \mathrm{d}s_{n-1} \cdots \int_{\xi}^{x} \varphi(s_1)\psi(s_1)\mathrm{d}s_1
$$
$$
= \left( \int_{\xi}^{x} \varphi(s)\psi(s)\mathrm{d}s \right)^{n-1}.
\tag{167}
$$

Therefore,

$$
I_{\xi,x} = \frac{1}{(n-1)!} \left( \int_{\xi}^{x} \varphi(s)\psi(s)\mathrm{d}s \right)^{n-1}
\tag{168}
$$

holds. Substituting this into the equation (163) yields the conclusion. $\qquad\square$

Next, we prove a functional identity needed for the proof of Proposition G.4.

**Lemma G.1.**

$$
I_\alpha(x) = \sum_{k=0}^{\infty} \frac{1}{k!\Gamma(k+\alpha+1)} \left( \frac{x}{2} \right)^{2k+\alpha}
\tag{169}
$$

*That is, let $I_\alpha$ be the modified Bessel function. Then, for $z \in \mathbb{C} \setminus \{0\}$, we have*

$$
H(z) := \sum_{k=0}^{\infty} \frac{z^k}{k!(k+1)!} = \frac{I_1(2\sqrt{z})}{\sqrt{z}}
\tag{170}
$$

*where*

$$
\sqrt{z} = \sqrt{|z|} e^{\frac{i}{2}\arg z}.
\tag{171}
$$

*Proof.* A direct calculation gives the following.

$$
\begin{aligned}
\sum_{k=0}^{\infty} \frac{z^k}{k!(k+1)!} &= \sum_{k=0}^{\infty} \frac{1}{k!(k+1)!} \left( \frac{2\sqrt{z}}{2} \right)^{2k} \\
&= \sum_{k=0}^{\infty} \frac{1}{k!(k+1)!\sqrt{z}} \left( \frac{2\sqrt{z}}{2} \right)^{2k+1} \\
&= \frac{I_1(2\sqrt{z})}{\sqrt{z}}
\end{aligned}
\tag{172}
$$

$\qquad\square$

**Remark G.2.** *Note that since $H(|z|) \le \exp(|z|)$, the radius of convergence of $H(z)$ is $\infty$.*

Using these, we can compute the exponential of the Separable Volterra Integral Operator $\mathcal{K}_{\varphi,\psi}$.

**Lemma G.3.** *For $c \in \mathbb{C}$,*

$$\exp\left(c\mathcal{K}_{\varphi,\psi}\right)[g](x) = g(x) + \varphi(x)\int_0^x g(\xi)\psi(\xi)\frac{cI_1\left(2\sqrt{c\int_\xi^x \varphi(s)\psi(s)\mathrm{d}s}\right)}{\sqrt{c\int_\xi^x \varphi(s)\psi(s)\mathrm{d}s}}\mathrm{d}\xi \tag{173}$$

*Proof.* By Lemma 6.5,

$$\begin{aligned}
\exp\left(c\mathcal{K}_{\varphi,\psi}\right)[g](x) &= \sum_{k=0}^\infty \frac{\left(c\mathcal{K}_{\varphi,\psi}\right)^k[g](x)}{k!} \\
&= g(x) + \sum_{k=1}^\infty \frac{\varphi(x)c^k}{k!\,(k-1)!}\int_0^x g(\xi)\psi(\xi)\left(\int_\xi^x \varphi(s)\psi(s)\mathrm{d}s\right)^{k-1}\mathrm{d}\xi \\
&= g(x) + \varphi(x)\int_0^x g(\xi)\psi(\xi)\left(\sum_{k=1}^\infty \frac{c^k}{k!\,(k-1)!}\left(\int_\xi^x \varphi(s)\psi(s)\mathrm{d}s\right)^{k-1}\right)\mathrm{d}\xi \\
&= g(x) + \varphi(x)\int_0^x g(\xi)\psi(\xi)\left(\sum_{k=0}^\infty \frac{c^{k+1}}{(k+1)!k!}\left(\int_\xi^x \varphi(s)\psi(s)\mathrm{d}s\right)^{k}\right)\mathrm{d}\xi \\
&= g(x) + \varphi(x)\int_0^x g(\xi)\psi(\xi)\left(c\sum_{k=0}^\infty \frac{1}{(k+1)!k!}\left(c\int_\xi^x \varphi(s)\psi(s)\mathrm{d}s\right)^{k}\right)\mathrm{d}\xi \\
&= g(x) + \varphi(x)\int_0^x g(\xi)\psi(\xi)cH\left(c\int_\xi^x \varphi(s)\psi(s)\mathrm{d}s\right)\mathrm{d}\xi \\
&= g(x) + \varphi(x)\int_0^x g(\xi)\psi(\xi)\frac{cI_1\left(2\sqrt{c\int_\xi^x \varphi(s)\psi(s)\mathrm{d}s}\right)}{\sqrt{c\int_\xi^x \varphi(s)\psi(s)\mathrm{d}s}}\mathrm{d}\xi
\end{aligned} \tag{174}$$

Here, the interchange of the integral and the series follows from the dominated convergence theorem, noting that $H(z)$ is continuous around $z = 0$. $\qquad\square$

In particular, this lemma allows us to compute the exponential of parametric CHIP.

**Proposition G.4.** *For $t \in \mathbb{R}_{>0}$ and $x \in [0, N]$,*

$$\exp\left(t\mathcal{F}_{\varphi,\psi,\omega}\right)[g](x) = e^{t\omega}\left(g(x) + \varphi(x)\int_0^x g(\xi)\psi(\xi)\frac{tI_1\left(2\sqrt{t\int_\xi^x \varphi(s)\psi(s)\mathrm{d}s}\right)}{\sqrt{t\int_\xi^x \varphi(s)\psi(s)\mathrm{d}s}}\mathrm{d}\xi\right) \tag{175}$$

*holds.*

*Proof.* By definition, we can write as

$$t\mathcal{F}_{\varphi,\psi,\omega} = t\mathcal{K}_{\varphi,\psi} + t\omega\mathcal{I}. \tag{176}$$

Since the operator $t\omega\mathcal{I}$ is a scalar multiple of the identity map, it commutes as an operator with $t\mathcal{K}_{\varphi,\psi}$, and hence we obtain

$$\exp\left(t\mathcal{F}_{\varphi,\psi,\omega}\right) = \exp\left(t\mathcal{K}_{\varphi,\psi}\right)\exp\left(t\omega\mathcal{I}\right). \tag{177}$$

From Lemma G.3:

$$\exp\left(t\mathcal{K}_{\varphi,\psi}\right)[g](x) = g(x) + \varphi(x)\int_0^x g(\xi)\psi(\xi)\frac{tI_1\left(2\sqrt{t\int_\xi^x \varphi(s)\psi(s)\mathrm{d}s}\right)}{\sqrt{t\int_\xi^x \varphi(s)\psi(s)\mathrm{d}s}}\mathrm{d}\xi, \tag{178}$$

and moreover

$$\exp\left(t\omega\mathcal{I}\right)[g] = e^{t\omega}g, \tag{179}$$

as a conclusion, we obtain

$$
\begin{aligned}
\exp\left(t\mathcal{F}_{\varphi,\psi,\omega}\right)[g](x) &= \exp\left(t\mathcal{K}_{\varphi,\psi}\right)\left[\exp\left(t\omega\mathcal{I}\right)[g]\right](x) \\
&= \exp\left(t\mathcal{K}_{\varphi,\psi}\right)\left[e^{t\omega}g\right](x) \\
&= e^{t\omega}\left(g(x) + \varphi(x)\int_0^x g(\xi)\psi(\xi)\frac{tI_1\left(2\sqrt{t\int_\xi^x \varphi(s)\psi(s)\mathrm{d}s}\right)}{\sqrt{t\int_\xi^x \varphi(s)\psi(s)\mathrm{d}s}}\mathrm{d}\xi\right).
\end{aligned}
\tag{180}
$$

$\square$

Using this Proposition G.4, we can prove Theorem 6.4 and Theorem 6.6.

*Proof of Theorem 6.4.* Since $\mathcal{C}$ is a bounded linear functional, there exists some $c \in L^2([0,N])$ such that

$$\mathcal{C}[f] = \int_0^N c(x)f(x)\mathrm{d}x. \tag{181}$$

Computing the convolution kernel $k_{\varphi,\psi,\omega,\chi}(t) = \mathcal{C}\exp\left(t\mathcal{F}_{\varphi,\psi,\omega}\right)\mathcal{G}_\chi[1]$ using Proposition G.4, we obtain

$$
\begin{aligned}
\exp\left(t\mathcal{F}_{\varphi,\psi,\omega}\right)\mathcal{G}_\chi[1](x) &= \exp\left(t\mathcal{F}_{\varphi,\psi,\omega}\right)[\chi](x) \\
&= e^{t\omega}\left(\chi(x) + \varphi(x)\int_0^x \chi(\xi)\psi(\xi)\frac{tI_1\left(2\sqrt{t\int_\xi^x \varphi(s)\psi(s)\mathrm{d}s}\right)}{\sqrt{t\int_\xi^x \varphi(s)\psi(s)\mathrm{d}s}}\mathrm{d}\xi\right),
\end{aligned}
\tag{182}
$$

and therefore

$$
\begin{aligned}
&\mathcal{C}\exp\left(t\mathcal{F}_{\varphi,\psi,\omega}\right)\mathcal{G}_\chi[1](x) \\
&= \int_0^N c(x)e^{t\omega}\left(\chi(x) + \varphi(x)\int_0^x \chi(\xi)\psi(\xi)\frac{tI_1\left(2\sqrt{t\int_\xi^x \varphi(s)\psi(s)\mathrm{d}s}\right)}{\sqrt{t\int_\xi^x \varphi(s)\psi(s)\mathrm{d}s}}\mathrm{d}\xi\right)\mathrm{d}x \\
&= e^{t\omega}\int_0^N c(x)\chi(x)\mathrm{d}x + e^{t\omega}\int_0^N\int_0^x c(x)\varphi(x)\chi(\xi)\psi(\xi)\frac{tI_1\left(2\sqrt{t\int_\xi^x \varphi(s)\psi(s)\mathrm{d}s}\right)}{\sqrt{t\int_\xi^x \varphi(s)\psi(s)\mathrm{d}s}}\mathrm{d}\xi\mathrm{d}x.
\end{aligned}
\tag{183}
$$

$\square$

*Proof of Theorem 6.6.* We obtain an explicit expression in the same manner as Theorem 6.4, and by substituting $\varphi = i\chi, \psi = i\overline{\chi}$, it can be written in the following form.

$$
\begin{aligned}
\exp\left(t\mathcal{F}_{i\chi,i\overline{\chi},\omega}\right)\mathcal{G}_\chi[1](x) &= e^{t\omega}\left(\chi(x) - \chi(x)\int_0^x \overline{\chi(\xi)}\chi(\xi)\frac{tI_1\left(2\sqrt{-t\int_\xi^x \overline{\chi(s)}\chi(s)\mathrm{d}s}\right)}{\sqrt{-t\int_\xi^x \overline{\chi(s)}\chi(s)\mathrm{d}s}}\mathrm{d}\xi\right) \\
&= e^{t\omega}\left(\chi(x) - \chi(x)\int_0^x |\chi(s)|^2\frac{tI_1\left(2i\sqrt{t\int_\xi^x |\chi(s)|^2\mathrm{d}s}\right)}{i\sqrt{t\int_\xi^x |\chi(s)|^2\mathrm{d}s}}\mathrm{d}\xi\right) \\
&= e^{t\omega}\left(\chi(x) + \chi(x)\int_0^x |\chi(\xi)|^2\frac{i\sqrt{t}I_1\left(2i\sqrt{t\int_\xi^x |\chi(s)|^2\mathrm{d}s}\right)}{\sqrt{\int_\xi^x |\chi(s)|^2\mathrm{d}s}}\mathrm{d}\xi\right)
\end{aligned}
\tag{184}
$$

By rewriting using the relation between Bessel and modified Bessel functions $iI_1(iz) = -J_1(z)$, we obtain the following.

$$\exp\left(t\mathcal{F}_{i\chi,i\overline{\chi},\omega}\right)\mathcal{G}_\chi[1](x) = e^{t\omega}\left(\chi(x) - \chi(x)\int_0^x |\chi(\xi)|^2 \frac{\sqrt{t}J_1\left(2\sqrt{t\int_\xi^x |\chi(s)|^2\,\mathrm{d}s}\right)}{\sqrt{\int_\xi^x |\chi(s)|^2\,\mathrm{d}s}}\,\mathrm{d}\xi\right) \tag{185}$$

Here, making the change of variables $y(\xi) = 2\sqrt{t}\sqrt{\int_\xi^x |\chi(s)|^2\,\mathrm{d}s}$, noting

$$\mathrm{d}y = -\frac{\sqrt{t}\,|\chi(\xi)|^2}{\sqrt{\int_\xi^x |\chi(s)|^2\,\mathrm{d}s}}\,\mathrm{d}\xi, \tag{186}$$

we obtain

$$\begin{aligned}
&e^{t\omega}\left(\chi(x) - \chi(x)\int_0^x |\chi(\xi)|^2 \frac{\sqrt{t}J_1\left(2\sqrt{t}\sqrt{\int_\xi^x |\chi(s)|^2\,\mathrm{d}s}\right)}{\sqrt{\int_\xi^x |\chi(s)|^2\,\mathrm{d}s}}\,\mathrm{d}\xi\right) \\
&= e^{t\omega}\left(\chi(x) + \chi(x)\int_{2\sqrt{t}\sqrt{\int_0^x |\chi(s)|^2\,\mathrm{d}s}}^0 J_1(y)\,\mathrm{d}y\right) \\
&= e^{t\omega}\left(\chi(x) - \chi(x)\int_0^{2\sqrt{t}\sqrt{\int_0^x |\chi(s)|^2\,\mathrm{d}s}} J_1(y)\,\mathrm{d}y\right).
\end{aligned} \tag{187}$$

In particular, using the differentiation formula for Bessel functions

$$\frac{dJ_0(y)}{dy} = -J_1(y), \tag{188}$$

we have

$$\begin{aligned}
\exp\left(t\mathcal{F}_{i\chi,i\overline{\chi},\omega}\right)\mathcal{G}_\chi[1](x) &= e^{t\omega}\left(\chi(x) + \chi(x)\int_0^{2\sqrt{t}\sqrt{\int_0^x |\chi(s)|^2\,\mathrm{d}s}} J_1(y)\,\mathrm{d}y\right) \\
&= e^{t\omega}\left(\chi(x) + \chi(x)\left(J_0\left(2\sqrt{t}\sqrt{\int_0^x |\chi(s)|^2\,\mathrm{d}s}\right) - J_0(0)\right)\right) \\
&= e^{t\omega}\left(\chi(x) + \chi(x)\left(J_0\left(2\sqrt{t}\sqrt{\int_0^x |\chi(s)|^2\,\mathrm{d}s}\right) - 1\right)\right) \\
&= e^{t\omega}\chi(x)J_0\left(2\sqrt{t}\sqrt{\int_0^x |\chi(s)|^2\,\mathrm{d}s}\right)
\end{aligned} \tag{189}$$

Thus, the integral kernel can be written explicitly as follows.

$$k_{i\chi,i\overline{\chi},\omega,\chi}(t) = \mathcal{C}\exp\left(t\mathcal{F}_{i\chi,i\overline{\chi},\omega}\right)\mathcal{G}_\chi[1] = e^{t\omega}\int_0^N c(x)\chi(x)J_0\left(2\sqrt{t}\sqrt{\int_0^x |\chi(s)|^2\,\mathrm{d}s}\right)\mathrm{d}x \tag{190}$$

$\square$

This immediately implies Theorem 6.1, the explicit-solution formula.

*Proof of Theorem 6.1.* In the formula of Theorem 6.6, by substituting $\chi(x) = \sqrt{2x+1}, \omega = -\frac{1}{2}$, the convolution kernel becomes

$$k_{i\chi,i\overline{\chi},\omega,\chi}(t) = e^{-\frac{1}{2}t}\int_0^N c(x)\sqrt{2x+1}J_0\left(2\sqrt{t}\sqrt{x^2+x}\right)\mathrm{d}x. \tag{191}$$

Thus the output $\Phi(v)$ is obtained by convolving with the above kernel: that is,

$$\Phi(v)(t) = \int_0^t \int_0^N c(x) e^{-\frac{1}{2}(t-\tau)} \sqrt{2x+1} J_0\left(2\sqrt{t-\tau}\sqrt{x^2+x}\right) v(\tau) \mathrm{d}x \mathrm{d}\tau \tag{192}$$

follows. $\qquad\square$

**Remark G.5.** *Note that in the above proof, even without assuming the continuity of $\chi \in L^2([0, N])$, the derivative in* (186) *can be represented in the same way almost everywhere, and hence the final equality still holds.*

