# OpenReview forum: "State Space Model with Continuous Limit of HiPPO Matrix: Eigenvalue Analysis and Explicit Solution Formula"
_ICML.cc/2026/Conference — ICML 2026 regular_

### Official Review · Reviewer_r6QC · 2026-03-11

**Soundness:** 3
**Presentation:** 3
**Significance:** 2
**Originality:** 3
**Overall Recommendation:** 4
**Confidence:** 3

**Summary:**

Structured State Space Models (SSSM) are deep learning architectures that have been recently proposed and that are very promising. The High-order Polynomial Projection Operators (HiPPO) matrix is a key element to achieve faster computation. Gu et al. 2022, the authors have provided a conjecture regarding the asymptotic behavior ot the eigenvalues of the normal part of the Hippo matrices. In this paper, the authors have introduce a Continuized-HiPPO (CHIP) operator and have analyzed the asymptotic behavior of the eigenvalues (theorems 5.4 and 5.5  and proofs in Appendix E and F). They have also demonstrated that the Linear State Space Layer (LSSL) induced by the CHIP operator admits an explicit integral representation.

**Compliance With Llm Reviewing Policy:**

Affirmed.

**Key Questions For Authors:**

1) What can be the practical use of the CHIP operator for SSSM ? A discussion on this topic is welcome.

2) What is the interest are presenting figures 1 to 4 ?

3) The overall quality of Appendix A  can be significantly improved.  The training of the parameters A and B are not explained in Algorithm 3.

**Limitations:**

yes

**Strengths And Weaknesses:**

Soundness : The submission sounds good. All theoretical results are well demonstrated.

Presentation: The paper is  well structured even if the paper is rather difficult to read.

Significance: The authors introduce the so-called CHIP operator and using it, they analyze the asymptotic behavior of eigenvalues of HiPPO matrices. The influence of this paper in future research or applications seems however unclear and perhaps limited.
Given the fact that the paper is pure applied mathematics,  the choice of the conference ICML is questionable.

Originality:  This proposed work is original. The authors have used advanced mathematical  tools (Banach, Hilbert spaces, Volterra integral operator, …) to introduce CHIP operator and derive the obtained results.

---

### Official Review · Reviewer_dD3C · 2026-03-12

**Soundness:** 3
**Presentation:** 2
**Significance:** 2
**Originality:** 3
**Overall Recommendation:** 3
**Confidence:** 2

**Summary:**

This paper studies the HiPPO matrix used in Linear State Space Layers and introduces a continuous-limit formulation called the Continuized-HiPPO operator, defined as a Volterra-type integral operator on an $L^2$ function space.

Using this operator perspective, the authors prove a previously stated conjecture on the asymptotic behavior of the eigenvalues of the normal part of the HiPPO matrix, deriving both leading-order and second-order terms. They also obtain an explicit integral expression for the corresponding state-space convolution kernel in terms of Bessel functions.

**Compliance With Llm Reviewing Policy:**

Affirmed.

**Key Questions For Authors:**

1. Can you clarify how the proven eigenvalue asymptotics (including the second-order correction term) concretely influence the design, initialization, stability, or performance of HiPPO-based SSM models in practice?
If you can demonstrate that this spectral characterization leads to improved initialization schemes, stability guarantees, or measurable training benefits, this would significantly strengthen the paper’s ML relevance and could positively affect my evaluation.

2. Beyond theoretical elegance, does the explicit Bessel-function kernel formulation provide any computational, numerical, or architectural advantages compared to existing S4-style structured kernels?
For example, does it reduce computational complexity, improve stability, simplify implementation, or enable new parameterizations? Clear evidence of such benefits, either theoretical or empirical, would materially improve the paper’s positioning for an ML venue.

3. The empirical evaluation is limited to Sequential MNIST with minimal reporting details (were multiple random seeds tested? standard deviations?) More importantly, I would have liked to see a broader evaluation of the method on more challenging long-range (e.g. LRA [1,2]) or modern sequence modeling benchmarks (e.g. ETT dataset from [3,4,5]). Why authors did not experimented on these well-established dataset?


4. Is the continuous operator formulation essential for obtaining the spectral results, or could similar asymptotic conclusions be derived directly from discrete matrix analysis?
Clarifying whether the operator viewpoint yields genuinely new insights (rather than being primarily a technical proof device) would help justify the conceptual contribution of the paper.

References

[1] Tay, Yi, et al. "Long range arena: A benchmark for efficient transformers." ICLR 2021

[2] Gu, Albert, Karan Goel, and Christopher Ré. "Efficiently modeling long sequences with structured state spaces." ICLR 2022

[3] Liu, Yong, et al. "itransformer: Inverted transformers are effective for time series forecasting." ICLR 2024

[4] Nie, Yuqi, et al. "A time series is worth 64 words: Long-term forecasting with transformers." ICLR 2023

[5] Wu, Haixu, et al. "Autoformer: Decomposition transformers with auto-correlation for long-term series forecasting." NeurIPS 2021

**Limitations:**

yes

**Strengths And Weaknesses:**

Strengths

- Rigorous and ambitious mathematical development.

- Nontrivial proof of an open conjecture.

- Elegant explicit kernel formulas.

- Careful spectral analysis with second-order asymptotics.



Weaknesses

- Weak motivation and ML positioning.
While the mathematical contribution is substantial, the connection to core ML questions is not sufficiently articulated. The paper does not clearly explain how the eigenvalue asymptotics or the continuous operator formulation materially affect model design, training stability, initialization strategies, or empirical performance in modern SSM architectures. As a result, the work reads primarily as a theoretical study of a specific matrix structure rather than a contribution that advances practical ML methodology.

- Writing and structure need major improvement.
The writing quality is a significant weakness.
The abstract is short and vague.
The Contributions section does not clearly state the resolved conjecture in plain terms.
Some definitions (e.g., the SSM kernel K(t)) are introduced without sufficient clarity.
Several passages give the impression of insufficient polishing.

- Minimal empirical validation.
There is no experimental evaluation at all. The only benchmark is sequential MNIST, reported in appendix, and it is poorly tested (e.g. a single run without standard deviation or description of the model selection). Were multiple seeds tested? Is the equivalence robust under hyperparameter variation? Without stronger empirical validation, the practical relevance remains weak.
More importantly, sequential MNIST is a standard but relatively simple benchmark.
If the goal is to demonstrate that the explicit-kernel formulation is practically competitive, broader benchmarks (e.g., long-range modeling tasks, language modeling, or time-series forecasting) would be expected.

- Lack of discussion of implications, and unclear practical gain from explicit solution.
The manuscript is largely a sequence of theoretical results about a specific SSM configuration based on HiPPO matrices. Even assuming full mathematical correctness, the practical implications for machine learning are not sufficiently developed.
It remains unclear: what new modeling capabilities the explicit kernel enables, whether it leads to computational advantages, or how it influences architectural design beyond theoretical reinterpretation.
If the primary contribution is the proof of an eigenvalue asymptotic formula and an explicit operator representation, the work may be more naturally positioned in a mathematically oriented journal rather than a premier ML conference, unless the ML impact is clarified more convincingly.

I'm giving a weak reject to acknowledge the good mathematical work, but because of the lack of ML positioning, empirical validation and alignment with the venue’s scope, it would deserve a plain reject in my opinion.

---

### Official Review · Reviewer_SNQh · 2026-03-13

**Soundness:** 3
**Presentation:** 3
**Significance:** 2
**Originality:** 4
**Overall Recommendation:** 3
**Confidence:** 1

**Summary:**

This paper studies HiPPO-based state space models. It uses the Continuized-HiPPO (CHIP) operator to analyze LSSL. The spectral properties of the normal part of the HiPPO matrix have been conjectured, but not formally proved. This paper fills in this gap.

**Compliance With Llm Reviewing Policy:**

Affirmed.

**Final Justification:**

My main concerns and questions are not addressed because there is no rebuttal.

**Key Questions For Authors:**

When will the continuous approximation be expected to be accurate or inaccurate?

**Limitations:**

I feel the paper might benefit from
- clarifying when the continuous approximation becomes inaccurate
- additional discussion about the applicability of theoretical results to recent SSMs.

**Strengths And Weaknesses:**

- This is a mathematically solid paper that has good contributions, such as the CHIP operator as a continuous-limit analogue of the HiPPO matrix, a formal derivation of the spectral properties, and an explicit solution for LSSL. It also makes a nice connection between SSM and tools from functional analysis.
- The results are mostly asymptotic or analytical. I feel the paper can be improved if it elaborates more about how the insights can be used to improve model design, learning, and computation.
- The models analyzed in this paper are simpler than the SSMs that are currently in use. While the simplification seems reasonable for the analysis to be feasible, the paper might benefit from elaborating more about whether this insight applies to those models.
- More comprehensive experiments might improve the paper.

---

### Official Review · Reviewer_kEHd · 2026-03-13

**Soundness:** 3
**Presentation:** 3
**Significance:** 3
**Originality:** 4
**Overall Recommendation:** 5
**Confidence:** 3

**Summary:**

This paper proposed the CHIP Operator as continuous limit of HiPPO matrix, and proved a conjecture regarding the asymptotic behavior of the eigenvalues of the HiPPO matrix's normal part (Albert Gu, 2022), a result previously supported only by numerical observations. It also provides an explicit solution formula for the LSSL equation.  The presentation of the paper is clear.

However, the proofs in this paper heavily rely on advanced mathematical concepts, such as Volterra integral operators, semigroup theory, which I am not familiar with. Consequently, I am unable to verify the formal correctness of the proofs.

**Compliance With Llm Reviewing Policy:**

Affirmed.

**Key Questions For Authors:**

1.The paper have proved the conjecture regarding the eigenvalue distribution of the HiPPO matrix's normal part. From a practitioner's perspective, how does this specific $\Theta(n^{-1})$ decay rate of eigenvalues directly influence the gradient stability or the memory horizon during the training of an LSSL? Does this theoretical result suggest a new way to initialize or regularize the state transition matrix $A$?

2.Modern state-of-the-art SSMs, such as Mamba, incorporate selective mechanisms where the transition matrices are input-dependent (time-varying). The current analysis is focused on the Linear Time-Invariant (LTI) setting. Do you see a path to extending the "Continuized-HiPPO" framework to non-LTI or selective SSMs,

**Limitations:**

The limitations of their experimental validation, which is primarily focused on verifying mathematical theorems rather than performance on standard AI benchmarks.

**Strengths And Weaknesses:**

Strengths：
1. The paper treating the HiPPO matrix as an integral operator on function spaces. This approach provides a new perspective for addressing the spectral analysis for discrete HiPPO matrices.
2. Through spectral analysis of the continuous operator, the paper provides a strict mathematical proof for the HiPPO eigenvalue distribution, filling a significant theoretical gap in the field.
3. The paper derived  an explicit solution for the SSM convolution kernel , which may facilitates the development of more efficient algorithms.

Weaknesses：
1. While the theoretical contribution is outstanding, the current experiments are primarily focused on verifying theoretical results (e.g., Sequential MNIST) and lack broad comparisons with other state-of-the-art models (like Mamba variants) on large-scale NLP tasks.
2. The theoretical results are heavily tied to the specific structure of the HiPPO-LegS matrix. While this is the most common variant, the modern SSM landscape has shifted toward generalized or learned transition matrices (like those in S4, S4D, or Mamba).

---

### Decision · Program_Chairs · 2026-04-30

**Decision:**

Accept (regular)

**Comment:**

First of all let me transcribe here a message I sent to the PC and SAC on April 9th (since there was no response, I did not take any action).

**Dear SAC and PC,

It looks like the authors of this paper mistakenly used the "Author AC Confidential Comment" feature to respond to the reviewers. This means that the latter never got to see their responses.

I am sorry to have overlooked this until now. Am I allowed to share the authors' responses with the reviewers?

Best,

AC**

Having said this, this paper contributes to the analysis of so-called state space models. The contribution is an asymptotic analysis of properties of the so-called HiPPO which is used in the initialization of these methods. The main finding is quite nice: an explicit description of an operator that is the large-size limit of this matrix. On the other hand, the main results are somewhat tangential to the practice (and even to much of the theory) of ML: there's no direct "killer application" that would justify this paper's existence. To my mind, this explains the disagreement between reviewers: they each "vote" according to how important applications and empirical evaluations are. My own opinion is that this is a nice paper, with nice technical idea that might turn out to be useful elsewhere. I thus recommend that it be accepted, although this is not a super-strong recommendation.